# CryoEM structural exploration of catalytically active enzyme pyruvate carboxylase

Jorge Pedro López-Alonso [1,4], Melisa Lázaro[1], David Gil-Cartón[1,2,4], Philip H. Choi[3], Alexandra Dodu[1], Liang Tong [3] & Mikel Valle [1] ✉

Pyruvate carboxylase (PC) is a tetrameric enzyme that contains two active sites per subunit that catalyze two consecutive reactions. A mobile domain with an attached prosthetic biotin links both reactions, an initial biotin carboxylation and the subsequent carboxyl transfer to pyruvate substrate to produce oxaloacetate. Reaction sites are at long distance, and there are several co-factors that play as allosteric regulators. Here, using cryoEM we explore the structure of active PC tetramers focusing on active sites and on the conformational space of the oligomers. The results capture the mobile domain at both active sites and expose catalytic steps of both reactions at high resolution, allowing the identification of substrates and products. The analysis of catalytically active PC tetramers reveals the role of certain motions during enzyme functioning, and the structural changes in the presence of additional cofactors expose the mechanism for allosteric regulation.

The number of structures of biomolecules determined by single particle cryo-electron microscopy (cryo-EM) is increasing exponentially in recent years[1]. The improvements in the hardware of microscopes and the software used for data analysis have resulted in more detailed maps that frequently arrive to the atomic-level resolution. Cryo-EM maps may also serve to identify multiple conformational states of complexes and their subunits[2,3]. The structural heterogeneity in particles can be discrete or continuous and can be separated by 3D classification. The different classes may account for low energy states in thermodynamic equilibrium and describe the functional space accessible to the protein[4]. This functional space is a limited part of the conformational space which can be modulated with the presence of allosteric regulators. The question is how far cryoEM can go in deciphering the functional structures of oligomeric complexes involved in multi-pathway reactions. This work explores by cryo-EM the functional space for such type of oligomers for the case of a biotin-dependent carboxylase.

Biotin-dependent carboxylases are metabolic enzymes that catalyze the transfer of carboxyl groups to different substrates[5,6]. Depending on the identity of carboxyl acceptor the enzyme can be pyruvate carboxylase (PC), propionyl-CoA carboxylase (PCC), acetyl-CoA carboxylase (ACC) and methylcrotonyl-CoA carboxylase (MCC). These enzymes are crucial in metabolic processes such as the fatty acid synthesis, gluconeogenesis and amino acid catabolism. In addition, there are two biotin-carboxylases only found in bacteria, geranyl-CoA carboxylase (GCC) and urea carboxylase (UC). Many of the UCs can also play as guanidine carboxylases[7]. Biotin-dependent carboxylases are usually active in homo- or hetero-oligomeric forms and perform their activity in two sequential steps at two different active sites[5,8]. In order to complete the reaction, the biotin co-factor must access the biotin carboxylase (BC) domain where it becomes carboxylated using MgATP and bicarbonate[8,9]. The carboxylated biotin is then translocated to the carboxyl transferase (CT) domain where the carboxyl group is transferred to the specific substrate. The biotin prosthetic group is attached to the biotin-carboxyl carrier protein (BCCP) domain that couples both reactions by traveling long distances between catalytic centers. Most of the published crystal structures show BCCP domains far from the catalytic centers or missing due to their flexibility[10]. The interaction of BCCP with CT domain has been described in PCC[11], ACC[12] and PC[13,14]. However, the interaction between BCCP and BC was only described in the mutant T882A from *Rhizobium etli* PC (RePC) and contains biotin in a position far from the active site[15]. The only structure of biotin in the catalytic position of BC corresponds

[1]CIC bioGUNE, Basque Research & Technology Alliance (BRTA), Bizkaia Technology Park, Derio, Bizkaia, Spain. [2]IKERBASQUE, Basque Foundation for Science, Bilbao, Spain. [3]Department of Biological Sciences, Columbia University, New York, NY, USA. [4]Present address: Basque Resource for Electron Microscopy, Instituto Biofisika (CSIC - UPV/EHU), Leioa, Spain. ✉e-mail: mvalle@cicbiogune.es

to the Biotin Carboxylase component from the ACC complex and was obtained by co-crystallization of the protein with free biotin[16]. Thus, a catalytically relevant structure for the interaction between the mobile BCCP and the BC domain is still missing.

Pyruvate carboxylase (PC) is a biotin-dependent, mitochondrial protein that catalyzes the carboxylation of pyruvate into oxaloacetate. This metabolite is essential in the tricarboxylic acid cycle[17] and fuels several anabolic reactions as gluconeogenesis, liponeogenesis, insulin secretion and synthesis of glutamate neurotransmitter[18]. In recent years, the central role of PC as a metabolic hub has been associated with reprograming and flexibility of cancer cells metabolism[19].

In addition to BC, CT and BCCP, an extra domain was found in crystallographic structures of PC (Fig. 1a)[14,20]. This domain links the other three functional ones and has been named as PC tetramerization domain (PT) due to its contribution to the oligomerization of the enzyme, or allosteric domain (AL), as it is essential to bind the allosteric activator acetyl-CoA[21,22]. PC is mostly found as tetramers of four identical subunits, with monomers of ~120–130 kDa in size[18]. This enzyme possesses a tetrahedral geometry with the tetramer organized in two layers[14,20,23]. In each layer, the monomer pairs run antiparallel to one another and perpendicular between the two faces. The monomer pairs of each face have minimal contacts with each other, and the tetramer is stabilized at each corner through BC-BC and CT-CT homodimer interactions. The BCCP domain is translocated from the BC of its own monomer to the CT domain of the opposite subunit in the same layer[20,23,24]. This pathway seems to be favored by the allosteric regulator acetyl-CoA, but the movement of each BCCP domain can follow different pathways, reaching the four active sites in its own layer[25]. Kinetic analysis of PC has suggested that both reactions occurring in the active sites of the BC and CT domains are well coordinated[23]. However, the communication mechanism to trigger the translocation of the BCCP domain between active sites in biotin-dependent carboxylases is not yet resolved.

In this work, we study the structure of *Lactococcus lactis* PC (LlPC) during catalysis in the presence of positive allosteric regulator acetyl-CoA. Using cryo-EM and unsupervised classification techniques, we determine conformations related with PC enzymatic activity. In addition, we analyze a LlPC sample containing the inhibitor cyclic di-3′,5′-adenosine monophosphate (c-di-AMP). Comparison of the different structures reveals insights into the pathway of the reaction and the mechanisms of the opposite allosteric regulations.

## Results

We obtained a cryo-EM map of LlPC with acetyl-CoA under reaction conditions with an estimated global resolution of 2.12 Å (Fig. 1b). This

resolution should allow the identification of side chains, the substrates and products of the reactions, and even ions and water molecules. However, there are large differences in resolution throughout the protein as observed in local resolution estimates. This initial map is a mixture of the different conformations of the enzyme during catalysis, and hence, BCCP domains are not visible at the current density threshold (Fig. 1c). After symmetry expansion of our dataset (see Methods), the first step of our analysis is the separation of catalytic states at both BC and CT reaction sites of all the subunits.

### BC domain

The BC domain is arranged into an ATP-grasp folding and contains four subdomains (A, residues 204-333; B, residues 131-203; C1, residues 1-130; and C2, residues 334-458). The B-subdomain is known to be very flexible and oscillates between closed and open states as a molecular lid over the ATP binding site[26], and in the cryoEM map appears as a disordered density. The absence of a strong signal for the BBCP region prompted us to perform a classification with a soft mask around the BC domain of the particles aligned in the framework of the full tetramer (Supplementary Fig. 1). This unsupervised classification produced a class with 9.4% of the particles showing good signal for the BCCP domain fully engaged in the BC reaction site ($BC_{react}$) with a closed B-lid (Supplementary Fig. 1b and Fig. 2a). In this class, the loop connecting BCCP and PT domains within the same subunit is well-defined (labeled with an asterisk in Fig. 2a). The BC-lid is closed grabbing the MgATP (Fig. 2b) where the adenine of ATP interacts with the main chain of Lys200 and Ile202, and the side chains of Glu199 and Lys157. The ribose has interactions with the side chains of His207, Gln231 and Asn234. The polyphosphate chain interacts with the amino group of Lys116, the $Mg^{2+}$ ion, and residues 161-164 in the T-loop. This $Mg^{2+}$ also coordinates the carboxylate groups of Glu274 and Glu286. Although at lower occupancy, there is a density consistent with a second cation coordinating the ATP γ-phosphate (asterisks in Fig. 2b, c) which has been proposed to be a second $Mg^{2+}$ ion[27]. Bicarbonate is placed between side chains of Arg290 and Glu294, and the main chain of Val293. At lower density values there is clear signal for the prosthetic biotin (Fig. 2c, d) in a position very similar to the observed for free biotin in *E. coli* Biotin Carboxylase[16] (Fig. 2e) and apart from the one observed in the mutant T882A from RePC[15] (Fig. 2f).

The rest of the segregated classes displayed large tilt movements of the BC domain itself. For this reason, the entire population was re-aligned with a refinement focused on this domain. A new classification with a soft mask around the BC domain of the re-aligned set of particles produced several classes with better defined movements of the BC region (Supplementary Fig. 1c): a class with the B-subdomain open,

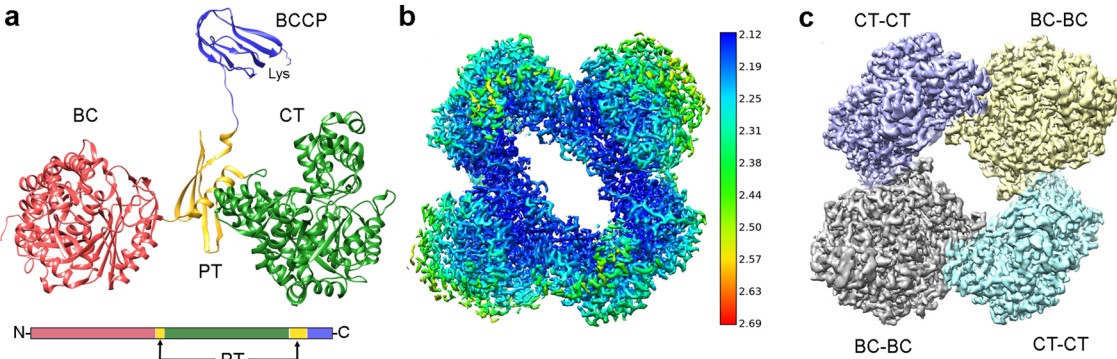

**Fig. 1 | Structure of LlPC. a** Atomic structure of LlPC (pdb code 5vyz[30]) subunit colored by different domains, together with the distribution of these domains in the primary structure of the protein. The lysine at the BCCP domain is the residue where prosthetic biotin is attached. **b** CryoEM map for LlPC in the presence of acetyl-CoA, substrates and co-factors. The 3D map is colored by estimates of local resolution, being the overall resolution 2.12 Å. **c** Rendering of the cryoEM map for LlPC depicted in b showing the BC-BC and CT-CT dimers from different layers. Abbreviations stand for: BC biotin carboxylase domain; CT carboxyl transferase domain; BCCP biotin-carboxyl carrier protein domain; PT PC tetramerization domain. This notation is used along all the figures.

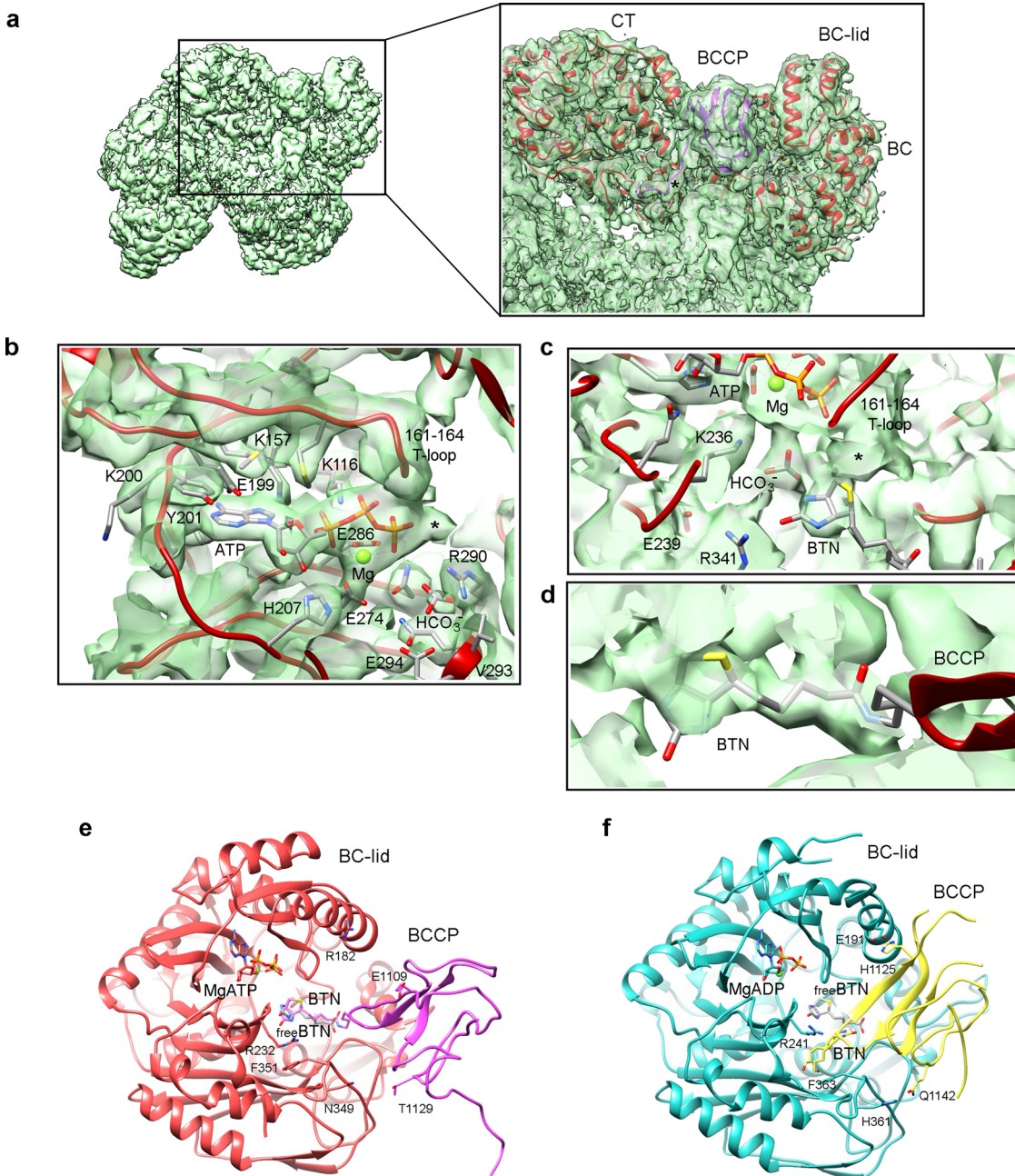

**Fig. 2 | Structural details of the BCCP-BC interaction. a** CryoEM map for the BC$_{react}$ class after classification. The atomic model for one LlPC subunit is seen in red ribbons but the BCCP domain that belongs to the same subunit is colored purple for clarity. Asterisk in zoom panel labels the connection between BCCP and PT domains. **b** Interactions of the BC domain with MgATP. Asterisk locates a putative second Mg$^{2+}$ ion. **c** Biotin binding site in the map for BC$_{react}$. BTN stands for biotin. **d** Close-up of biotin density shown in the cryoEM map at 3.5σ threshold. **e** Interaction of LlPC BCCP with BC domain in the BC$_{react}$ class comparing the position of the biotin with the free biotin in *E. coli* BC (pdb code 3G8C). **f** Interaction observed in RePC T882A (pdb code 3TW6) of BCCP (yellow) with BC domain (blue) also including the free biotin observed in *E. coli* BC. The BCCP domain of RePC binds in a different position to the one of LlPC and its biotin is far from the active site. The tethered biotin of LlPC lays in a position similar to the free biotin of *E. coli* BC. BTN stands for biotin here and in following figures.

BC$_{open}$; two classes with the B-subdomain closed, BC$_{closed}$; and six classes showing the BCCP domain oriented towards the BC active site, BC$_{BCCP}$.

Comparing the maps for BC$_{open}$ and BC$_{closed}$ states (Fig. 3) it is seen that the B-subdomain lid in BC$_{open}$ opens around 50° (Fig. 3a, and Supplementary Movie M1) what could allow the release of ADP and the entrance of new substrate. The residues forming the hinge of the molecular lid are Ile131 and Glu203. Arg171 forms a new salt bridge with Glu203 that could stabilize the open conformation of the B-subdomain lid (Fig. 3b). There is a strong density for ATP in the BC$_{closed}$ state (rendered at 2.5 σ in Fig. 3g), but the map for the BC$_{open}$

class shows a weaker density in the nucleotide binding site (visible at 1.5 σ in Fig. 3f) compatible with partial release of the nucleotide upon opening of the B-subdomain lid. We attribute this density to unreleased ADP molecules present in some of the processed particles. Essentially, the interaction of Lys116 with the β-phosphate of ATP in the BC$_{closed}$ state (Fig. 3g) seems to be conserved in the BC$_{open}$ class between the same amino acid and the last phosphate of the putative ADP molecule (Fig. 3f). In this configuration, the ADP would interact only with the amine of Lys116 and the main chain of Ile202 at both ends of the molecule, and the side interactions are missing. By comparison of the interaction networks in the BC$_{open}$ (Fig. 3b, d) and

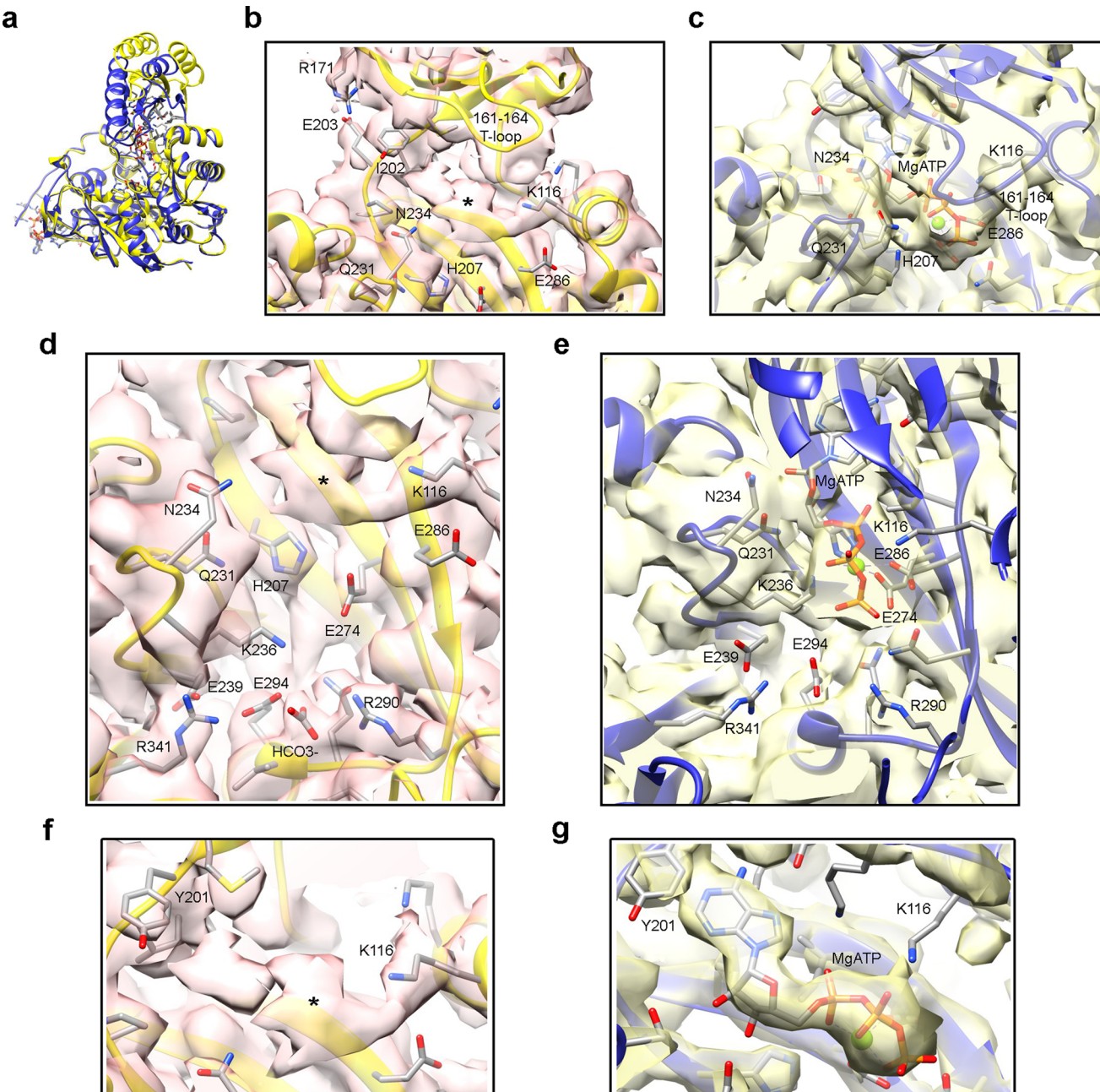

**Fig. 3 | Open and closed B-subdomain lid. a** Atomic models for the $BC_{open}$ (yellow) and $BC_{closed}$ (blue) BC-subdomain lids at the BC site. **b** and **d** BC catalytic site at the $BC_{open}$ state. **c** and **e** BC catalytic center with bound MgATP in the $BC_{closed}$ state. **f** and **g** Close-up views of the densities at the nucleotide binding site for $BC_{open}$ (**f**) and $BC_{closed}$ (**g**) rendered at 1.5 σ and 2.5 σ density thresholds respectively. Along the panels, the density attributable to bound nucleotide in the $BC_{open}$ class is labeled with asterisks (see main text).

in the $BC_{closed}$ (Fig. 3c, e), it can be seen that the interactions of MgATP with Glu274, Glu286, His207, Gln231, Lys120 and the T-loop are lost upon the opening of the B-subdomain lid. The missing interactions in the $BC_{open}$ between the nucleotide and the 231–239 loop liberate this region of the BC domain and Glu239 can interact with Arg341 through a salt bridge (Fig. 3d). In the presence of MgATP, this loop binds to the ribose of the nucleotide, the density of the salt bridge between Glu239 and Arg341 weakens, and the density of Arg341 changes its orientation and cannot accommodate the full side chain of the amino acid, suggesting that Arg341 is more free from the interaction and that it can map the configuration required to receive an incoming biotin. This salt bridge between Arg341 and Glu239 has been reported to be produced only after ATP cleavage to avoid

the reentry of carboxybiotin into the active site and prevent its decarboxylation[28].

In summary, the cryoEM maps accurately describe the motion of the ATP-grasp module and the configuration of the BC active site with its substrates and co-factors that goes along with the catalytic mechanism proposed for RePC[28] where: Glu294 and Arg290 promote the deprotonation of bicarbonate for the nucleophilic attack on the γ-phosphate of MgATP; the resulting carboxyphosphate decompose to $CO_2$ and the $PO_4^{3-}$ that deprotonates the biotin at $N_1$; and Arg341 plays a central role on the stabilization of deprotonated biotin to accept the $CO_2$ (Fig. 4).

In the classification of the BC region more than 50% of the monomers have BCCP in the proximity of BC (classes $BC_{BCCP}$ in

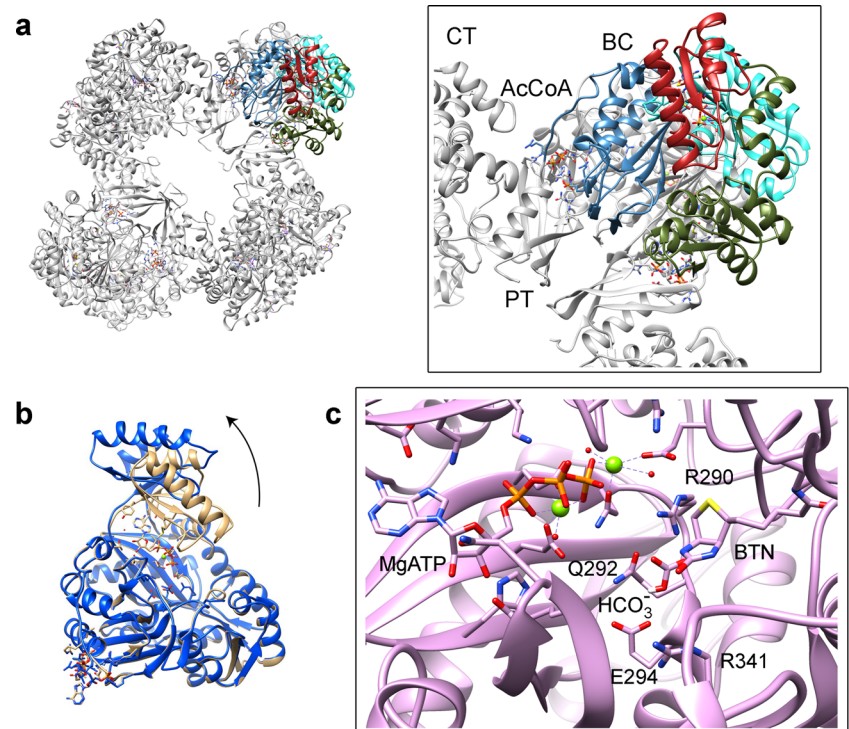

**Fig. 4 | Atomic models for the BC domain during catalytic reaction. a** Model of tetrameric LlPC and enlargement of a BC domain. The A sub-domain is colored in cyan, B-subdomain in red, C1-subdomain in green, and C2-subdomain in blue. The B-subdomain interacts mainly with ATP and most of the catalytic residues are located in the A-subdomain. **b** Overlap of BC_open (blue) and a BC_closed (brown) models. The arrow shows the movement of the B-subdomain (B-lid). **c** Active site of BC_react. The tethered biotin lays close to the HCO3- molecule which interacts with Glu294, Arg290 and Gln292. ATP is stabilized by two $Mg^{2+}$ ions. AcCoA stands for acetyl-CoA here and in following figures.

Supplementary Fig. 1). BCCP binds BC domain through a limited set of interactions that would explain its high mobility, and the presence of BCCP does not produce large conformational changes in BC domain as it was previously reported[29]. In all these maps for the BC_BCCP classes the definition for the BCCP is poor and the attached biotin cannot be located at the BC active site, suggesting a large structural heterogeneity.

## CT domain

A focused classification with a mask around the CT domain produced: a class with no substrate CT_empty; two classes with pyruvate CT_pyr; a class with BCCP inserted at the active site CT_react; and five classes with oxaloacetate CT_oxa (Supplementary Fig. 2). The cryoEM maps of these catalytic states show local densities that define the nature of the bound molecules with high definition (Fig. 5), and also large differences in the degree of openness of the CT domain, being very open in the CT_empty class, intermediate at variable degrees within CT_pyr and CT_oxa classes, and highly closed in the CT_react class with bound BCCP domain (left panels in Fig. 5 and Supplementary Movie M2).

The main catalytic residues of the CT domain are placed in a TIM barrel (residues 525-780) (Fig. 6a). This region is very well-resolved in the volumes, allowing the identification of most of the side chains, ligands and even some water molecules (Supplementary Fig. 3). Residues 781-1000 form 12 helices composing a funnel domain that leads to the active site (Fig. 6a). The active site contains a cation, which is coordinated to His732, His734, Arg533, Asp534, the two carboxylic atoms of carboxylysine 703 and a water molecule that also interacts with pyruvate O3. This ion has been identified as $Mn^{2+}$ for the crystallographic models of *S. aureus*[13], *L. lactis*[30], *L. monocytogenes*[31] and human[14] PC. On the other hand, the structure of PC from *R. etli*[20] contains a $Zn^{2+}$. We have assigned it as $Mn^{2+}$ in our models to be consistent with previous works.

The arrangement of the active site of CT agrees with that proposed by Sheng and coworkers[32].

The funnel subdomain of CT_react is more closed. The helix 827-840 closes approximately 16.2° with respect to the CT_empty class. A density for BCCP bound to the mobile region of the funnel subdomain is observed. This binding is stabilized by several interactions between the BCCP β-strand 1105–1111 and the loop 897–900 of the funnel sub-domain (Supplementary Fig 4b). There are clear densities for biotin at the active site and for oxaloacetate, and the arrangement of the BCCP and the biotin at the active site is very similar to structures previously reported for PC[13,14,33]. In this CT_react complex, the density of the loop connecting BCCP to the PT domain is blurred, so the structure cannot define whether the BCCP domain comes from the opposite or from the same subunit.

During the transition from CT_empty to CT_pyr or CT_oxa, the 825–935 region, a subdomain of the funnel, closes and moves closer to the active site (Fig. 6b). The loop 608–618 of the TIM barrel also moves with the funnel. In the CT_empty class without any substrate, the funnel subdomain is very open and Arg606 moves away from the active site, establishing an interaction with the side chain of Asn609 (Fig. 6b). The movement of this arginine was reported in the apo-RePC[34], and its mutation resulted in loss of PC activity[33].

The TIM barrel is bound to the PT domain through a long loop, followed by two helices leading to the interior of the TIM barrel. This region contains a cation bound to the main chains of residues Val519 and Thr522, and the side chain of Asp753 (Supplementary Fig. 3c). This ion has been tentatively assigned as $Mg^{2+}$ based on a similar assignment in RePC[20], however, its role is not clear[18].

## Interplay between reaction sites

After the classification and analysis of the BC and CT reaction centers we explored the relationship between different catalytic steps within

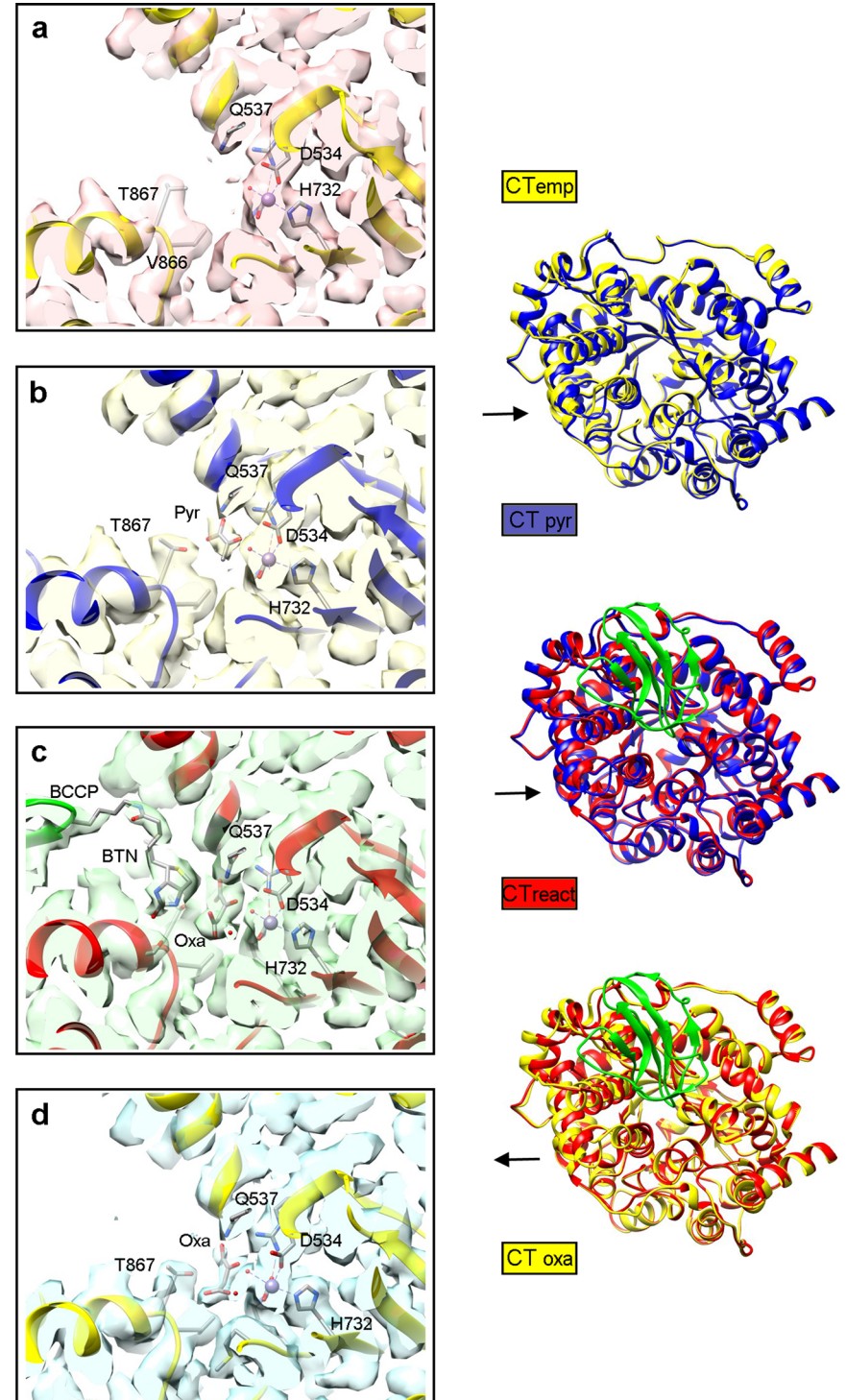

**Fig. 5 | Catalytic states at the CT active site.** CryoEM maps together with modeled atomic coordinates are shown focused on the active site of the CT domain. The four states displayed correspond to: **a** an empty active site (CT$_{empty}$); **b** pyruvate bound (CT$_{pyr}$); **c** biotin inside the active site and density for oxaloacetate (CT$_{react}$); **d** oxaloacetate at the active site (CT$_{oxa}$). On the right panels the pairs of the corresponding atomic models show the movement of the funnel sub-domain during closing and opening of the active site. Maps are rendered at 2σ density threshold. Abbreviations stand for: Pyr pyruvate; and Oxa oxaloacetate. This notation is used along the figures.

the tetramer. For each set of particles within the different classes we look at the distribution of the other subunits taking into account the rotational transformations during the symmetry expansion. In this way, we can record the increase or decrease of each class associated with a catalytic state in a specific position within the tetramer compared to the rest of the catalytic sites in the same or other subunits,

building a heat map. The deviations from the average along the compared subunits were small and in the range of ±10%, although most of the values are closer to ±5% (Fig. 7). The largest differences are observed for BC$_{react}$ state pairs that are enriched in subunits from different layers and bound by direct BC-BC contacts (Fig. 7d) but are reduced when the interlayer contact goes through CT-CT regions

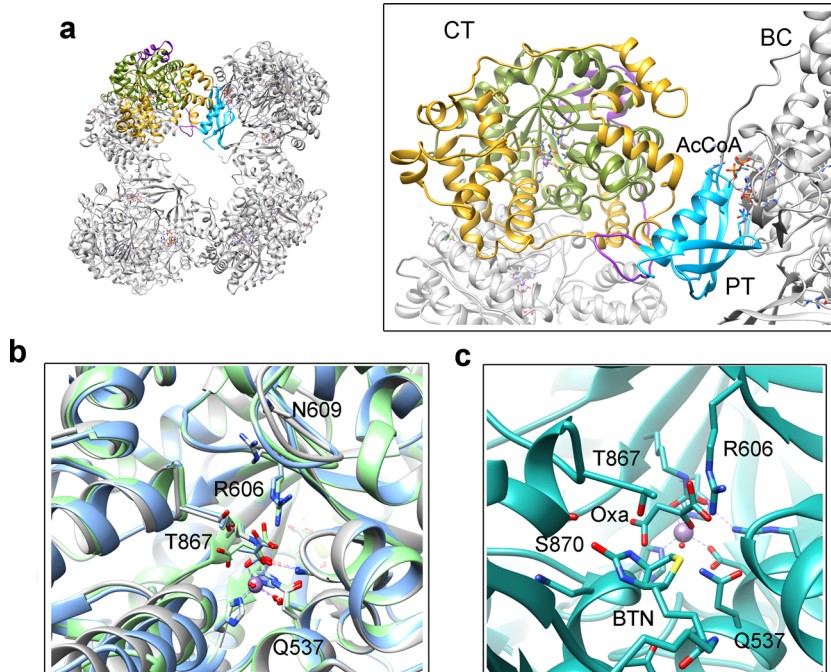

**Fig. 6 | Atomic details for CT and PT domains. a** Model of tetrameric LlPC and enlargement of CT and PT domains from one subunit. The PT domain is colored in cyan, and at the CT the TIM barrel in green, the funnel subdomain in yellow and the connecting loop in purple. The CT active site is located at the entrance of the TIM barrel and surrounded by the funnel domain. **b** Overlap of $CT_{empty}$ (gray), $CT_{pyr}$ (blue) and $CT_{oxa}$ (green) models. The largest conformational changes are produced in the helix 827-840 (bottom left corner) and the loop 610-617 (top). Arg606 interacts with pyruvate/oxaloacetate. In $CT_{empty}$ it moves to the outside and establishes a new interaction with Asn609. **c** In the $CT_{react}$ model biotin interacts with Ser870. The hydroxyl group of Thr867, the methyl group of pyruvate and the amide group of biotin interact and have an extra density between them produced by the carboxyl group. In the figure, this group is shown bound to pyruvate producing oxaloacetate.

(Fig. 7b). This $BC_{react}$ class also shows $BC_{open}$ enrichment and $BC_{closed}$ depletion when BC domains are in direct contact (Fig. 7d).

There are only a couple of patterns related to the CT domains: showing a positive correlation of the same catalytic state when they are in the same layer (Fig. 7c); and a negative one when they are in subunits from different layers bound by their CT-CT regions (Fig. 7b). A similar behavior is also shown by the $BC_{open}$ conformation which displays a positive correlation between subunits in the same layer and a negative one with catalytic states in subunits for different layers bound by their BC-BC regions (Fig. 7d).

Interestingly, there are no strong correlations between $BC_{BCCP}$ or $BC_{react}$ and $CT_{react}$ when they are in the same subunit (Fig. 7a) or in different subunits in the same layer (Fig. 7c) suggesting that BCCP can interact with the CT domain of both subunits from the same layer as previously suggested[25].

### Global movements

The separation of reaction steps at catalytic sites does not reveal the large conformational changes in the PC tetramer that could be coupled to the function of the enzyme. To explore the conformational space of the tetramers we performed a multibody refinement using masks enclosing dimers of BC-BC or CT-CT catalytic domains at different layers (as the bodies shown in Fig. 1c). These masks define regions that could move using the allosteric domain (and the binding site for acetyl-CoA) as the main hinge while keeping the interface between identical (BC-BC and CT-CT) domains in different layers unaltered. The results of this multibody refinement reveal that the first 10 eigenvectors account for 73% of the variance of the sample and no predominant movements are present (Supp. Figure 5). The main movement of the first vector corresponds to a tilt of the BC dimers ($BC_{tilt}$) where the BC domains of one layer move towards the protein center and the ones at the other layer move outwards (vector 1 in Fig. 8). The main movement

in the second vector is a twist motion (vector 2 in Fig. 8). This movement is identical in both layers and produces the approach of the two catalytic domains within the monomer ($BC-CT_{twist}$). The third vector showed a mix of motions but was very similar to the twist observed in the second one. The fourth vector produces an asymmetric $CT_{tilt}$ where the two CT domains in the same layer move in opposite directions (vector 4 in Fig. 8). When these vectors are compared with the classifications focused on catalytic centers, some correlations are observed for several classes. It seems that there is a slight correlation between the $BC_{tilt}$ vector with $CT_{empty}$ class as this class is favored by the movement of its BC toward the center of the protein. The strongest correlation is seen for the $CT_{react}$ and $CT_{empty}$ classes with the fourth vector, the $CT_{tilt}$ motion, and it seems that the $CT_{react}$ catalytic state is linked to the $CT_{tilt}$ towards the center of the protein. When particles from the ends of this vector are refined, the resulting maps show that the densities for BCCP domains are linked to the CT domains that tilt toward the center of the tetramer and absent near the CT domains that move away from the center of the protein (Fig. 9a). In this latter case the movement of the CT region away from the center of the tetramer places the CT active site out of reach for the BCCP domain of the opposite subunit (Fig. 9b) that would require a longer loop connected to the PT domain.

### Allosteric changes on LlPC

LlPC has a specific activity of 127 min$^{-1}$ which is slightly increased in the presence of the allosteric regulator acetyl-CoA to 154 min$^{-1}$ (Fig. 10a). The inhibitor c-di-AMP reduces LlPC activity to 50 min$^{-1}$. This reduced activity is recovered to 139 min$^{-1}$ by the addition of acetyl-CoA. In order to determine the structural changes behind these differences in enzymatic activity we compared the structure of LlPC in the presence and absence of acetyl-CoA and c-di-AMP. Since there are crystallographic models of the protein in complex with c-di-AMP and without

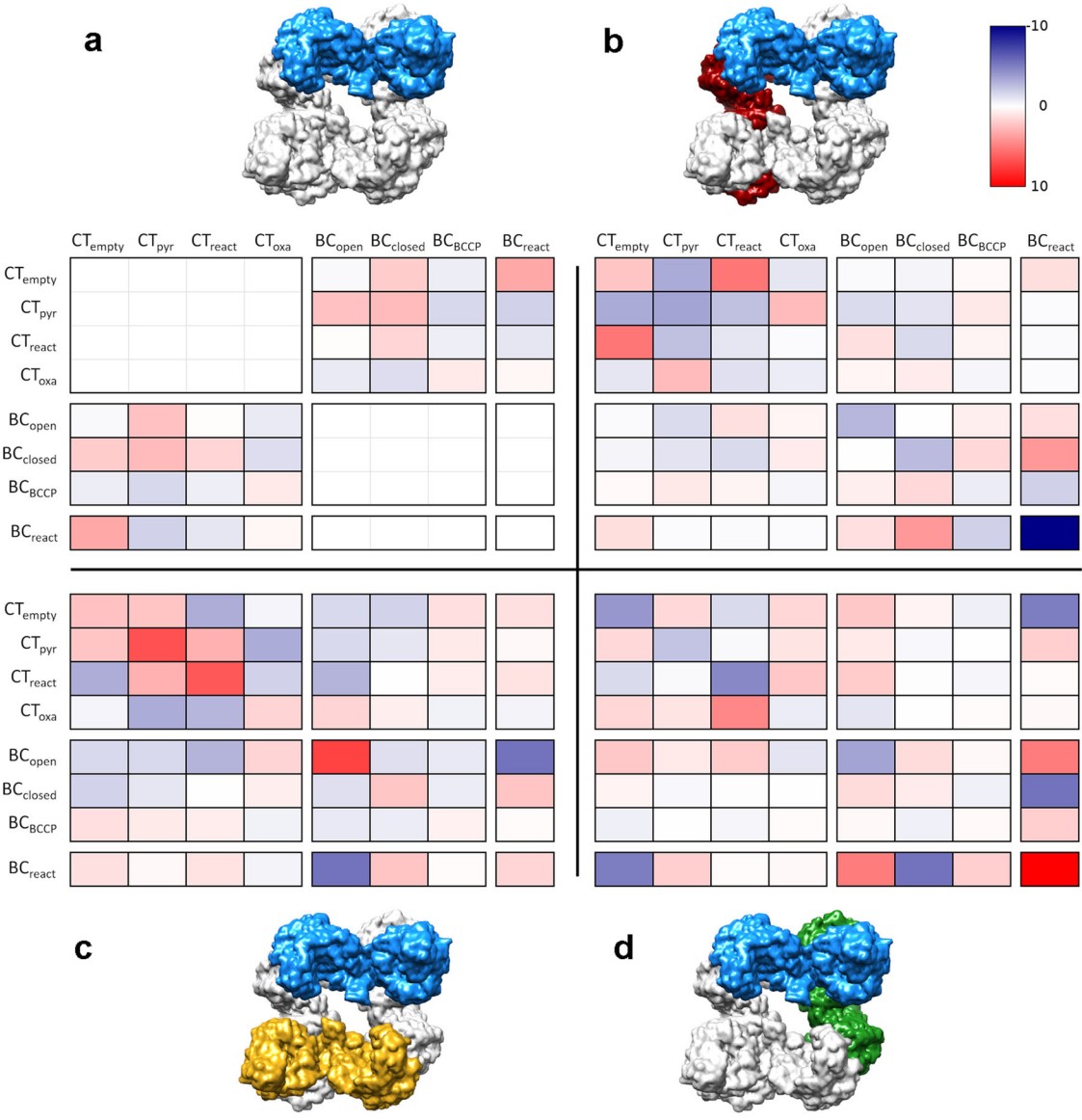

**Fig. 7 | Heat map of the correlations between the different classes after classification of BC and CT reaction centers.** Deviations from the average of the catalytic states compared to the other subunits. Red color shows a positive correlation (the correlation is larger than the average) and blue color a negative correlation (the correlation is lower than the average). **a** Correlations within a monomer (as showed in the model, in blue). The empty cells are the correlations between identical catalytic states in the same monomers. **b** Correlations between two monomers bound by the CT domains (blue and red monomers). **c** Correlations between domains in the same layer (blue and yellow monomers). **d** Correlations between two monomers bound by the BC domains (blue and green monomers). Color scale bar is seen next to **b**.

acetyl-CoA[30], we have explored the structure of LlPC with acetyl-CoA and in the presence/absence of c-di-AMP. Our previous described sample accounts for the LlPC complex in the presence of acetyl-CoA (Fig. 1b). We calculated a new cryoEM map at 3.37 Å of resolution for a sample including both acetyl-CoA and c-di-AMP in the mixture (Supplementary Fig. 6).

The LlPC structures, in the presence and absence of c-di-AMP, are very similar to the crystallographic models with no acetyl-CoA[30] being the main difference the angle between BC and CT domains where the allosteric domain plays as the hinge of the structural variations. The binding of the allosteric activator acetyl-CoA to this domain modifies the angle between BC and CT domains in 1.4° (Fig. 10b) while the angle between CTs remains unaltered (Fig. 10c). The inhibitor c-di-AMP binds at the interface between CT domains and changes their relative arrangement by 8.5°. This change is transferred across the tetramer and the angle between BC and CT changes in 9.7° in the opposite direction to the observed for acetyl-CoA. When acetyl-CoA is added to

the c-di-AMP sample, the activator recovers the angle between BC and CT domains (Fig. 10b) as well as the activity of the inhibited enzyme (Fig. 10a).

## Discussion

The aim of this work was to examine the conformations of LlPC and the communication between active sites during catalysis. Time-resolved cryo-EM has been used for studying the short-lived conformations produced during enzymatic reactions[35]. However, this technique is not suitable for our study as PC contains eight active sites and the tetramer might follow different reaction pathways. Since a monomer cannot be distinguished from the others, we are not able to follow the redistribution of the protein population into different conformations as time progresses. Our approach analyzes the structural landscape of PC tetramers during catalytic reactions with turnover of substrates and co-factors added in molar excess at the beginning of the sample preparation.

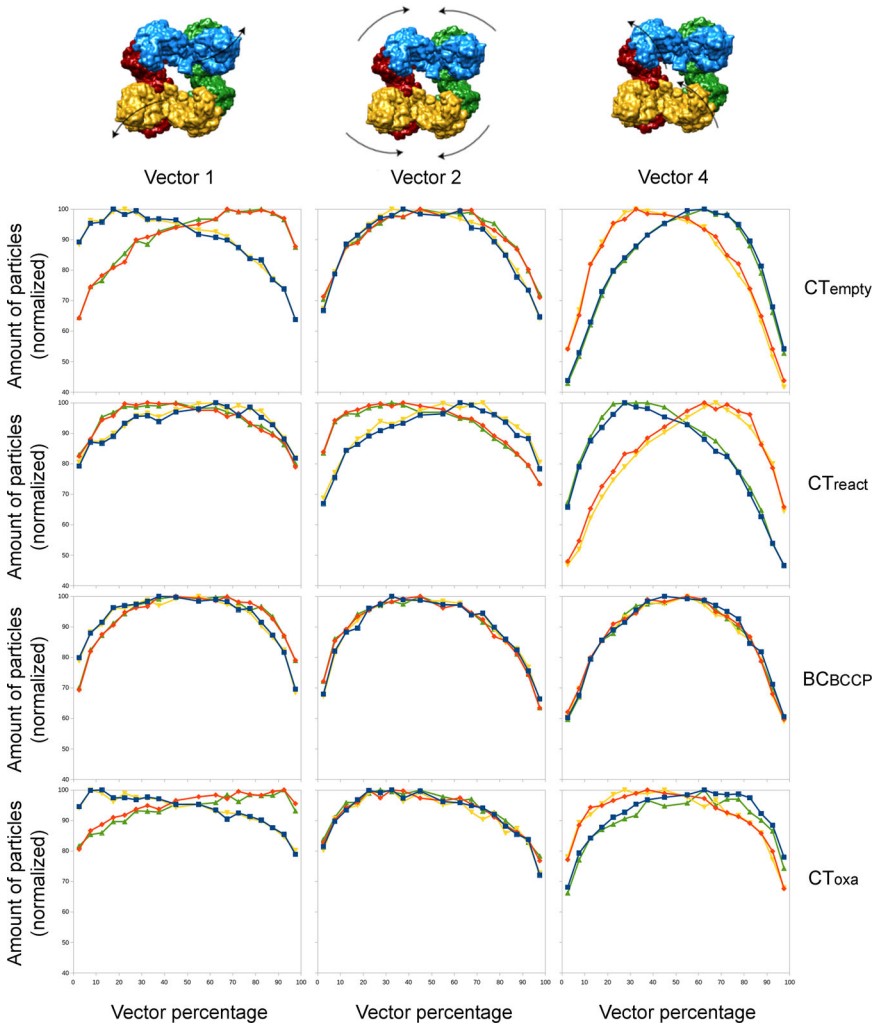

**Fig. 8 | Analysis of the correlations between global motions and selected BC and CT classes.** The columns show 3 main vectors of the multibody analysis and the rows show selected classes from BC and CT classification. Tetrameric models at the top display the different movements associated with the eigenvectors: $BC_{tilt}$ movement (vector 1), $BC$-$CT_{twist}$ movement (vector 2), and $CT_{tilt}$ movement (vector 4). The graphs show the normalized quantity of particles of a class in each vector percentage. The colors of the curves correspond to a PC monomer and match the colors at the top LlPC models.

From the mechanism point of view, BCCP domains couple both reactions of PC. Most of the crystal structures of PC lack signal for these domains due to their flexible nature, but they can be observed by cryoEM[36]. According to the classes resulting from focused classifications of LlPC catalytic centers, around 60% of BCCPs are nearby the active site of BC domain and 9% are inserted into the active site of the CT domain (Supplementary Figs. 1 and 2). This observation indicates that less than 30% of BCCP domains are not seen interacting with the catalytic centers, either because they are being translocated, or because they are in an inactive conformation. These percentages support the notion that the reaction in BC is the rate limiting step in PC functioning[37].

The position of biotin during BC reaction has been studied in *E. coli* acetyl-CoA carboxylase using free biotin and bicarbonate[16]. The interaction of BCCP and BC was only described in the mutant T882A from *R. etli* PC (RePC) but contains biotin in a position far from the active site[15] (Fig. 2f). In our cryoEM map for $BC_{react}$ state biotin is placed in a catalytic position very similar to the one described previously using free biotin (Fig. 2e). The interaction between BCCP domain and the B-subdomain lid of BC is stablished only when the lid is closed through a set of point contacts, and the large tip containing the prosthetic biotin has a wide range of positions available. This suggests that the position and orientation of the BCCP module in the vicinity of

the BC site does not constrain the binding of the biotin to a specific site but allows the biotin itself to look for a high affinity binding site. Moreover, the release of Arg341 from its salt bridge with Glu239 upon ATP binding (Fig. 3e) paves the way for the binding of the incoming biotin and regulates its access to the catalytic site.

In the same sample we have also found BCCP in the active site of CT domain, in a conformation very similar to the observed for human CT (HsPC)[14] and for *Staphylococcus aureus* PC (SaPC)[13,14,33]. The position of BCCP and the arrangement of the active site is similar in the three models as the interaction of BCCP with the funnel domain is very stable. $CT_{react}$ contains a density between the molecule of pyruvate and biotin. This density is probably due to a combination of the carboxyl group bound to biotin or bound to pyruvate.

In the $CT_{react}$ state the domain funnel closes upon interaction with the inserted BCCP. In this pose there is a large interacting surface between BCCP and CT regions (approximately twice the surface of interaction between BCCP and BC), and the position of the biotin at the tip of BCCP is directed toward the catalytic center. The closing of the CT funnel starts with the binding of pyruvate substrate and reaches its maximum when BCCP enters the site. It is known that the binding of pyruvate induces the formation of a binding pocket for carboxybiotin[34], and our results suggest that the presence of pyruvate also triggers the initial closure of the CT funnel and facilitates the

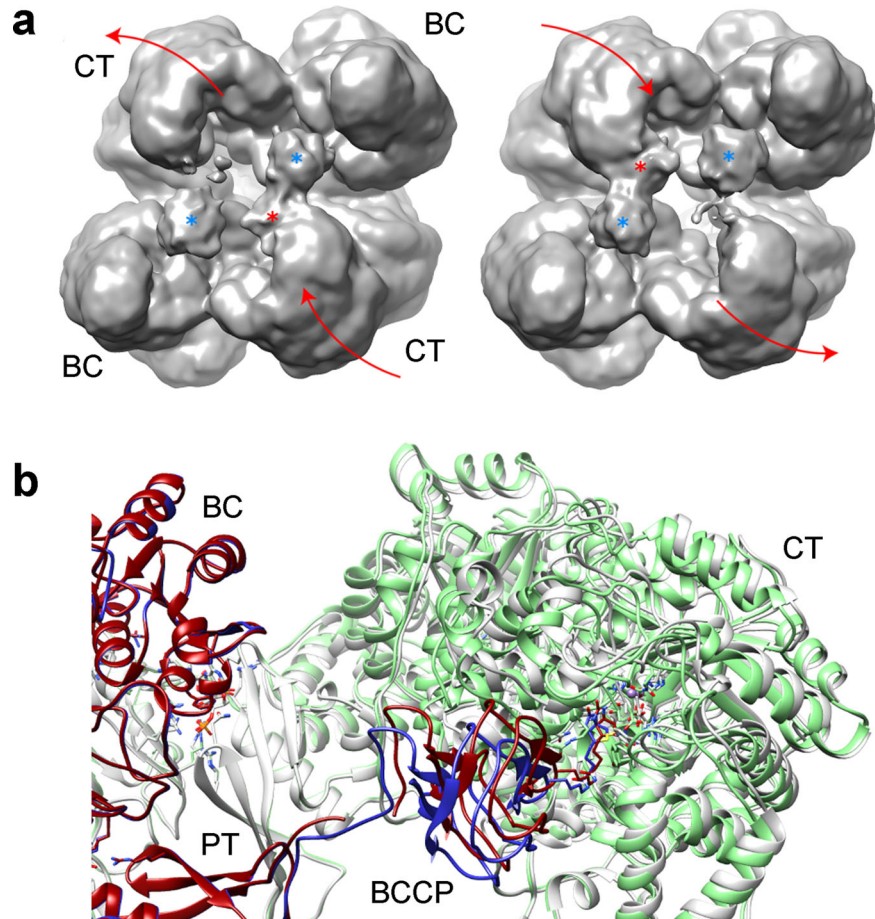

**Fig. 9 | Correlation between CT$_{\text{tilt}}$ movement and occupancy of the CT site by BCCP. a** Cryo-EM densities of LlPC obtained from 5% of particles at the beginning (left) or the end (right) of vector 4. The volumes have been filtered to improve the density for BCCP domains and they are rendered at 1.1σ of density threshold. Vector 4 corresponds to CT$_{\text{tilt}}$ and the movement has been highlighted with red arrows. BCCPs have been marked with asterisks, blue if they interact with BC domain or red if they interact with CT domain. **b** Overlay of the atomic models for two positions of CT$_{\text{tilt}}$ in gray and green. The models were fitted according to the BC of the opposite chain (colored in red/blue). The BCCP was placed in each CT domain according to the CT$_{\text{react}}$ model. The model in gray/red shows an 8 Å gap in the loop connecting BCCP and PT domains.

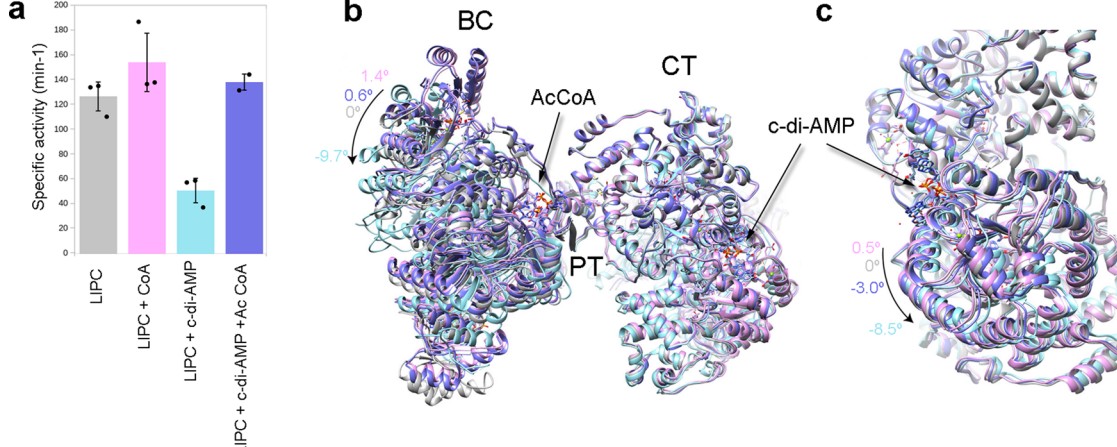

**Fig. 10 | Allosteric regulation and structural changes in LlPC. a** Specific activity of LlPC in the absence and presence of allosteric regulators acetyl-CoA and c-di-AMP. Data are presented as mean values (bars) and SD (error bars), and dots represent each independent assay with $n = 3$ for all the samples but LlPC with acetyl-CoA and c-di-AMP with $n = 2$. Source data are provided as a Source Data file. **b** Structural comparison of LlPC apo (pdb code 5VYW), with acetyl-CoA (current work), with c-di-AMP (pdb code 5VYZ) and with acetyl-CoA and c-di-AMP (current work). The models were aligned using the TIM barrel of CT as reference. The arrow highlights the movement of the BC domain and its degree of amplitude with respect to the apo protein. **c** Relative movement between the two CT domains in contact in the same set of structures. Along the figures, the color code is: LlPC apo gray; with acetyl-CoA pink; with c-di-AMP cyan; and with acetyl-CoA and c-di-AMP purple.

interaction with the BCCP domain carrying the carboxybiotin. Thus, the substrate-induced changes play at local (binding pocket for carboxybiotin) and domain (closure of the CT region) levels to facilitate the recruitment of the BCCP-carboxybiotin complex.

Thus, at both BC and CT active sites, the recruitment of the BCCP domain and its prosthetic biotin is regulated by changes in the active sites driven by the binding of substrates and cofactors and, presumably, by the different affinities that biotin or carboxybiotin display for the reaction centers. Questions remain as to whether conformational changes in the tetramer drive the movement of the BCCP domain to couple the BC and CT active sites, and the exact mechanism of the allosteric regulation.

We have studied the global movements of LlPC using multibody refinement. Our results indicate that PC tetramers display a combination of different motions (Fig. 8 and Supplementary Fig. 5) and that the study of correlations between multibody analysis and focused classifications on catalytic centers may help to understand complex dynamical processes as those produced by this oligomeric enzyme. In the current case, we have observed that BCCP cannot arrive at the CT domain when CT tilts away from the protein center (Fig. 9). This CT-tilt movement may help BCCP to reach the CT domain and to be released from it after the carboxyl transferase reaction takes place. During the reaction in CT, BCCP establishes several interactions between the strand 1105-1111 and the loop 897–900 in the funnel sub-domain (Supplementary Fig. 4b). The tilting movement of the CT domain may help to overcome the energy barrier required to break the interaction network between BCCP and the funnel subdomain.

The correlations between classes for BC and CT reaction sites of different subunits (Fig. 7) are modest and compatible with a multipathway reaction. We do not observe negative or positive correlations between $BC_{react}$ and $CT_{react}$ in the same subunit (Fig. 7a) or in the same layer (Fig. 7c), and the connection of the BCCP domain with the PT region is not well defined in the cryoEM map for $CT_{react}$. This suggests that the multi-pathway reaction involves BCCP domains that can interact with the CT domains of both subunits of the same layer[25] but only with the BC site of its own subunit.

Noteworthy, the highest positive correlation corresponds to the $BC_{react}$ pair joint by the BC-BC interaction surface (Fig. 7d), suggesting a cooperative behavior of BC regions in opposite layers to recruit biotin-carrying BCCP domains. This is also supported by the enrichment between $BC_{react}$ and $BC_{open}$, and by the depletion between $BC_{react}$ and $BC_{closed}$ when the BC domains are in direct contact (Fig. 7d). All in all, it seems that the binding of BCCP to the BC domain in one layer stimulates the recruitment of the BCCP to the opposite layer when ready ($BC_{closed}$). This is in disagreement with the postulated negative cooperativity for biotin carboxylase dimers in the so-called half-of-sites reactivity mode, where BC subunits communicate to alternate in their catalytic states[38,39]. This model is based on some observations of the alternate mode of action of the biotin carboxylase dimers of the ACC multienzyme complex[38]. In our data with tetrameric LlPC, a clear negative correlation is observed between the BC domains of the subunits bound by their CT regions for the $BC_{react}$ state (Fig. 7b). We have also observed that the PC subunits bound by their CT domains show a negative correlation in their catalytic state (Fig. 7b). The correlations between active sites and tetrameric motions are loose, suggesting that the movements of the BCCP domains linked by highly flexible loops are not tightly coupled to any of the conformational changes of the tetramer, but that some of the motions allow or favor them. Nevertheless, there is a significant coupling between the CT-tilt motion and an asymmetric catalytic state of the CT active site between layers (Fig. 8). To decipher whether it is a genuine negative cooperativity mediated by CT-CT contacts and the possible mechanism for its regulation, additional insights are needed.

In all the cryoEM maps for LlPC analyzed in this work, the subunits have a symmetrical organization without major differences in their overall architecture. This is relevant since an asymmetric configuration has been described in other PC tetrameric structures, such as in the crystallographic structures for RePC[15,20], and for Listeria monocytogenes PC (LmPC)[31], and in the low-resolution cryoEM structure for SaPC[23]. In the asymmetric configuration, the position of the PT domain is significantly different when subunits from different layers are compared and there is a rotation of about 60–70° between BC and CT regions. In the cryoEM work of SaPC engaged in catalytic activity[23], although at low resolution, the asymmetric tetramer was understood as a catalytically relevant state. Our current results for LlPC do not show any asymmetric architecture and most of the relevant stages of the enzymatic activity are in display. The asymmetric quaternary structure has been described in PC enzymes with low intrinsic activity that require the activation by acetyl-coA (as it is the case for RePC, SaPC, and LmPC), and in tetrameric organizations without the activator[31] or with only two molecules of acetyl-CoA (or ethyl-CoA) bound to each tetramer[15,20]. One possibility is that the asymmetric PC structure appears in tetramers that do not contain the four allosteric activator molecules that are needed to be fully competent. To clarify whether the asymmetric state has any enzymatic relevance, future studies are required.

Our results of the effect of the allosteric regulators in the structure of PC suggest that there is a relationship between the activity of PC and the angle between BC and CT domains of the subunits within the tetrameric arrangement. The effect of this angle in PC activity could be mediated by the different accessibility of BCCP to the CT domain as observed in multibody refinement analysis. We do not have access to the analysis of the motions that PC displays while bound to different allosteric modulators, but the clear correlation between the enzymatic activity and the angle that is shown in the hinge between the BC and CT domains (Fig. 10) suggests that allosterism plays by narrowing the spectrum of available tetrameric motions.

In summary, we have resolved several catalytic states of LlPC with the allosteric activator acetyl-CoA and the motions that LlPC tetramers undergo during catalysis. By crossing these analyses, we conclude that the BCCP domain with the flexible linker can move between active sites driven by the affinity of carboxylated or non-carboxylated biotin and the readiness of the catalytic regions to accept it. The movement of the BCCP domain is not tightly coupled to the conformational changes of the tetramer, but the motions of the oligomer can facilitate its access to the active sites, and allosteric regulators modulate the landscape of these motions, that is, they narrow the conformational space which in turn conditions the functional space of the enzyme Table 1.

## Methods

### Enzymatic assays

LlPC was expressed in *E. coli* and purified following protocols described earlier[30]. The catalytic activity of LlPC was determined at 10 °C by coupling the production of oxaloacetate to the oxidation of NADH by malate dehydrogenase[40]. The decreasing concentration of NADH was monitored by measuring the absorbance at 340 nm. The reaction mixture contained 20 mM Tris-HCl, (pH 7.6), 200 mM NaCl, 50 mM sodium bicarbonate, 20 mM pyruvate, 5 mM MgCl₂, 2 mM DTT, 2 mM ATP, 350 µM NADH, 15 U of malate dehydrogenase and 0.15 µM of LlPC. Some samples included 2 mM acetyl-CoA and/or 100 µM of c-di-AMP.

### Vitrification

LlPC was incubated in a solution containing 20 mM Tris-HCl (pH 7.6), 200 mM NaCl, 5 mM MgCl₂, 2 mM DTT and 2 mM acetyl-CoA at 37 °C for 2 min. Then, a solution of pyruvate in the same buffer was added and incubated on ice. Finally, a reaction buffer containing bicarbonate and ATP was added to the LlPC sample. After mixing, the sample was immediately applied to a Quantifoil grid R2/1 300 Mesh with carbon, incubated for 30 s and vitrified using a Vitrobot (FEI) set to 4 °C and

## Table 1 | Cryo-EM data collection, refinement and validation statistics

| | #1 LlPC + c-di-AMP (EMDB-15037) (PDB 7ZZ8) | #2 LlPC (EMDB-15028) PDB 7ZYY | #3 LlPC CT$_{oxa}$ (EMDB-15029) PDB 7ZYZ |
|---|---|---|---|
| **Data collection and processing** | | | |
| Magnification | 81,000 | 81,000 | 81,000 |
| Voltage (kV) | 300 | 300 | 300 |
| Electron exposure (e–/Å$^2$) | 95 | 48 | 48 |
| Defocus range (μm) | –0.9 to –5.1 | –0.2 to –3.0 | –0.2 to –3.0 |
| Pixel size (Å) | 1.085 | 1.06 | 1.06 |
| Symmetry imposed | C1 | C1 | C1 |
| Initial particle images (no.) | 2,168,776 | 4,578,464 | 1,369,092 |
| Final particle images (no.) | 2,090,256 | 1,369,092 | 120,548 |
| Map resolution (Å) | 3.29 | 2.12 | 2.97 |
| FSC threshold | 0.143 | 0.143 | 0.143 |
| Map resolution range (Å) | 3.05-4.5 | 2.12-3.37 | 2.39-6.22 |
| **Refinement** | | | |
| Initial model used (PDB code) | 5VYZ | 5VYW | 5VYW |
| Model resolution (Å) | 3.26 | 2.1 | 2.1 |
| FSC threshold | 0.143 | 0.143 | 0.143 |
| Model resolution range (Å) | 3.05-4.5 | 2.12-3-37 | 2.39-6.22 |
| Map sharpening B factor (Å$^2$) | –129 | –41 | –40 |
| Model composition | | | |
| Non-hydrogen atoms | 33,724 | 33,924 | 4,433 |
| Protein residues | 4,230 | 4,224 | 541 |
| Ligands | 26 | 28 | 3 |
| B factors (Å$^2$) | | | |
| Protein | 66.66 | 43.87 | 50.69 |
| Ligand | 76.95 | 47.50 | 58.19 |
| R.m.s. deviations | | | |
| Bond lengths (Å) | 0.004 | 0.004 | 0.004 |
| Bond angles (°) | 0.694 | 0.742 | 0.775 |
| Validation | | | |
| MolProbity score | 1.39 | 1.36 | 1.62 |
| Clashscore | 4.45 | 4.43 | 4.79 |
| Poor rotamers (%) | 0.11 | 1.22 | 2.18 |
| Ramachandran plot | | | |
| Favored (%) | 97.03 | 97.65 | 97.39 |
| Allowed (%) | 2.97 | 2.35 | 2.61 |
| Disallowed (%) | 0 | 0 | 0 |
| | #4 LlPC CT$_{empty}$ (EMDB-15030) (PDB 7ZZ0) | #5 LlPC CT$_{pyr}$ (EMDB-15032) (PDB 7ZZ2) | #6 LlPC CT$_{react}$ (EMDB-15031) (PDB 7ZZ1) |
| **Data collection and processing** | | | |
| Magnification | 81,000 | 81,000 | 81,000 |
| Voltage (kV) | 300 | 300 | 300 |
| Electron exposure (e–/Å$^2$) | 48 | 48 | 48 |
| Defocus range (μm) | –0.2 to –3.0 | –0.2 to –3.0 | –0.2 to –3.0 |
| Pixel size (Å) | 1.06 | 1.06 | 1.06 |
| Symmetry imposed | C1 | C1 | C1 |
| Initial particle images (no.) | 1,369,092 | 1,369,092 | 1,369,092 |

## Table 1 (continued) | Cryo-EM data collection, refinement and validation statistics

| | #1 LlPC + c-di-AMP (EMDB-15037) (PDB 7ZZ8) | #2 LlPC (EMDB-15028) PDB 7ZYY | #3 LlPC CT$_{oxa}$ (EMDB-15029) PDB 7ZYZ |
|---|---|---|---|
| Final particle images (no.) | 118,294 | 113,462 | 128,599 |
| Map resolution (Å) | 2.75 | 2.95 | 2.77 |
| FSC threshold | 0.143 | 0.143 | 0.143 |
| Map resolution range (Å) | 2.24-5.99 | 2.41-5.92 | 2.24-6.09 |
| **Refinement** | | | |
| Initial model used (PDB code) | 5VYW | 5VYW | 5VYW |
| Model resolution (Å) | 2.1 | 2.1 | 2.1 |
| FSC threshold | 0.143 | 0.143 | 0.143 |
| Model resolution range (Å) | 2.24-5.99 | 2.41-5.92 | 2.24-6.09 |
| Map sharpening B factor (Å$^2$) | –29 | –40 | –31 |
| Model composition | | | |
| Non-hydrogen atoms | 4,387 | 4,366 | 4,958 |
| Protein residues | 541 | 541 | 618 |
| Ligands | 2 | 3 | 4 |
| B factors (Å$^2$) | | | |
| Protein | 52.24 | 64.06 | 60.71 |
| Ligand | 67.10 | 81.11 | 69.93 |
| R.m.s. deviations | | | |
| Bond lengths (Å) | 0.004 | 0.004 | 0.005 |
| Bond angles (°) | 0.697 | 0.736 | 0.886 |
| Validation | | | |
| MolProbity score | 1.34 | 1.65 | 1.62 |
| Clashscore | 3.28 | 4.21 | 3.90 |
| Poor rotamers (%) | 1.96 | 3.27 | 2.68 |
| Ramachandran plot | | | |
| Favored (%) | 98.13 | 97.76 | 97.38 |
| Allowed (%) | 1.87 | 2.24 | 2.62 |
| Disallowed (%) | 0 | 0 | 0 |
| | #7 LlPC BC$_{react}$ (EMDB-15033) (PDB 7ZZ3) | #8 LlPC BC$_{open}$ (EMDB-15035) (PDB 7ZZ5) | #9 LlPC BC$_{closed}$ (EMDB-15034) (PDB 7ZZ4) |
| **Data collection and processing** | | | |
| Magnification | 81,000 | 81,000 | 81,000 |
| Voltage (kV) | 300 | 300 | 300 |
| Electron exposure (e–/Å$^2$) | 48 | 48 | 48 |
| Defocus range (μm) | –0.2 to –3.0 | –0.2 to –3.0 | –0.2 to –3.0 |
| Pixel size (Å) | 1.06 | 1.06 | 1.06 |
| Symmetry imposed | C1 | C1 | C1 |
| Initial particle images (no.) | 1,369,092 | 1,369,092 | 1,369,092 |
| Final particle images (no.) | 128,647 | 152,048 | 182,007 |
| Map resolution (Å) | 2.41 | 2.43 | 2.63 |
| FSC threshold | 0.143 | 0.143 | 0.143 |
| Map resolution range (Å) | 2.28–3.45 | 2.35-6.63 | 2.53-6.82 |
| **Refinement** | | | |
| Initial model used (PDB code) | 5VYW | 5VYW | 5VYW |
| Model resolution (Å) | 2.4 | 2.1 | 2.3 |

## Table 1 (continued) | Cryo-EM data collection, refinement and validation statistics

| | #1 LlPC + c-di-AMP (EMDB-15037) (PDB 7ZZ8) | #2 LlPC (EMDB-15028) PDB 7ZYY) | #3 LlPC CT$_{oxa}$ (EMDB-15029) (PDB 7ZYZ) |
|---|---|---|---|
| FSC threshold | 0.143 | 0.143 | 0.143 |
| Model resolution range (Å) | 2.28-3.45 | 2.35-6.63 | 2.53-6.82 |
| Map sharpening $B$ factor (Å$^2$) | −31 | −39 | −54 |
| **Model composition** | | | |
| Non-hydrogen atoms | 34,427 | 3,661 | 3,682 |
| Protein residues | 4,311 | 461 | 463 |
| Ligands | 26 | 3 | 3 |
| **$B$ factors (Å$^2$)** | | | |
| Protein | 66.19 | 70.67 | 78.25 |
| Ligand | 68.45 | 111.23 | 121.03 |
| **R.m.s. deviations** | | | |
| Bond lengths (Å) | 0.003 | 0.005 | 0.004 |
| Bond angles (°) | 0.717 | 0.818 | 0.794 |
| **Validation** | | | |
| MolProbity score | 1.53 | 1.67 | 1.47 |
| Clashscore | 4.89 | 4.95 | 5.07 |
| Poor rotamers (%) | 1.72 | 1.86 | 0.27 |
| **Ramachandran plot** | | | |
| Favored (%) | 97.51 | 96.73 | 96.75 |
| Allowed (%) | 2.49 | 3.27 | 3.25 |
| Disallowed (%) | 0 | 0 | 0 |

100% humidity. The final concentrations were 0.1 µM LlPC, 2 mM DTT, 2 mM acetyl-CoA, 20 mM pyruvate, 2 mM ATP, 10 mM KHCO$_3$, 5 mM MgCl$_2$, 200 mM NaCl, 20 mM Tris pH 7.6.

A second sample also containing 100 µM c-di-AMP was vitrified onto a Quantifoil grid R2/1 200 Mesh with carbon in the same conditions.

### Cryo-EM data acquisition and data processing

Automated data acquisition was carried out at the eBIC facilities (Diamond LightSource, UK) with a Titan Krios electron microscope (FEI) at 300 kV and a K3 direct detector (GATAN) operating in counting mode and using EPU software. In total 10,518 movies were collected each containing 40 frames over a 3.99 s exposure at 81,000X magnification, and yielding a pixel size of 1.06 Å. The total exposure was 48 electrons/Å$^2$.

Movie frames were aligned using MotionCor 2 1.2.6 using dose-weighting option[41] (representative cryoEM image of the sample is shown in Supplementary Fig. 7). Contrast transfer function (CTF) for each aligned micrograph was estimated using CTFfind 4.1.5[42]. A total of 4,578,464 particles were automatically selected using Laplacian autopick in RELION 3.06[43]. Several rounds of reference-free 2D class averaging and unsupervised 3D classification were used to clean the data until a final number of 342,273 particles.

An initial 3D refinement resulted in a resolution of 2.89 Å. This reconstruction was used to initiate several rounds of CTF refinement and Bayesian polishing[44] and the polished particles were employed for a new 3D refinement with D2 symmetry. To avoid the loss of resolution due to the conformational heterogeneity in every asymmetric unit of the protein, the symmetry was expanded (Briggs et al., 2005). Each particle of the data set was quadruplicated, and its Euler angles were modified so every asymmetric unit was aligned[45]. The information on the rotation to which each quadruplicated particle was subjected was stored. A final refinement using local searches yielded a resolution of

2.12 Å with a total of 1,369,092 particles. Local resolution was calculated with RELION.

An unsupervised 3D classification in RELION was performed using a soft mask around the BC domain. In addition, focused refinements of CT and BC domains were performed, followed by 3D classifications using soft masks around the respective domains without particle alignment. Every BC and CT class was refined using local searches. The volumes were B-factor sharpened with values of B-factor automatically calculated in RELION[46] and ranging from 30 to 60 Å$^2$.

Particle sets for each class were compared to each other to obtain correlations. The information of rotation of each quadruplicated particle provided additional information on which monomer correlates. Deviations of these correlations from the percentage of particles in each class allow the identification of co-occurrence of classes in different subunits of the protein.

The quadruplicated data were also refined using multibody refinement in RELION 3.1 using four rigid bodies[47]. Two bodies comprised two contacting BC domains from different layers and the corresponding allosteric domains. The other two bodies were composed of two contacting CT domains. The correlations of particles from different classes along these eigenvectors were also analyzed.

Data acquisition for LlPC with the inhibitor c-di-AMP was also carried out at the eBIC facilities with Titan Krios electron microscope (FEI) at 300 kV and a Falcon III direct detector. A total of 5869 movies were recorded each containing 39 frames over 1 s exposure at a magnification of 75,000X with a pixel size of 1.085 Å and a total exposure of 95 electrons/Å$^2$. Movie frames were aligned using MotionCor 2 and applying the dose-weighting option, and contrast transfer function (CTF) for each aligned micrograph was estimated using CTFfind 4. A total of 2,172,689 particles were automatically selected using Xmipp auto-picking function[48] in the Scipion framework[49]. Several rounds of reference-free 2D class averaging and unsupervised 3D classification in RELION were used to clean the data until a final number of 522,564 particles. An initial 3D refinement gave a resolution of 3.71 Å. The data were quadruplicated and refined. The new reconstruction was used to initiate several rounds of CTF refinement and Bayesian polishing, and the resulting particles were employed for a new 3D refinement yielding a resolution of 3.37 Å.

### Model building

The crystallographic model 5VYW[30] was used as a template for model building. The different domains were rigid body fitted to each density. The maps were sharpened by reference-based amplitude scaling using LocScale[50] in the CCPEM suite. Structures were optimized with several iterative cycles of real-space refinement in Phenix[51] using secondary structure and Ramachandran restraints followed by manual model building using Coot[52]. The figures were prepared using Chimera[53].

### Reporting summary

Further information on research design is available in the Nature Research Reporting Summary linked to this article.

## Data availability

The data that support this work are available from authors upon reasonable request. CryoEM maps generated in this study have been deposited in the Electron Microscopy Data Bank with IDs: EMD-15028 (LlPC overall map); EMD-15033 (BC$_{react}$); EMD-15035 (BC$_{open}$); EMD-15034 (BC$_{closed}$); EMD-15030 (CT$_{empty}$); EMD-15032 (CT$_{pyr}$); EMD-15031 (CT$_{react}$); EMD-15029 (CT$_{oxa}$); and EMD-15037 (LlPC with acetyl-CoA and c-di-AMP).

Generated atomic models have been deposited in the Protein Data Bank with codes: 7ZYY (LlPC overall map); 7ZZ3 (BC$_{react}$); 7ZZ5 (BC$_{open}$); 7ZZ4 (BC$_{closed}$); 7ZZ0 (CT$_{empty}$); 7ZZ2 (CT$_{pyr}$); 7ZZ1 (CT$_{react}$); 7ZYZ (CT$_{oxa}$); and 7ZZ8 (LlPC with acetyl-CoA and c-di-AMP). Source data are provided with this paper.

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

## Acknowledgements

We acknowledge Diamond Light Source for cryoEM data collection under proposals EM-15997 and BI-22006. This study was supported by grants from the HFSP (RGP0062), and from the Spanish Ministerio de Ciencia e Innovación (PGC2018-098996-B-100) to M.V., and grants from the NIGMS (R35GM118093) and the NIAID (R01AI116669) to L.T. We thank Aileen Santini for proofreading the manuscript.

## Author contributions

J.P.L., M.L., A.D. and P.H.C. produced protein samples and performed the biochemical assays. J.P.L. and D.G. prepared and screened the cryoEM grids. J.P.L. worked on the image processing of all the data sets with guidance by M.V. J.P.L., L.T., and M.V. designed the experiments, analyzed the results and wrote the paper.

## Competing interests

The authors declare no competing interests.
