## [Peer Review File · Nature Communications]

CryoEM structural exploration of catalytically active enzyme pyruvate carboxylaseReviewer #1 (Remarks to the Author):

The manuscript by López-Alonso et al describes a series of conformational states in pyruvate carboxylase from *Lactococcus lactis* captured by unclassification techniques applied to cryo EM. Using this approach, this team has captured important new conformational states for pyruvate carboxylase (most notably, a catalytically competent BCCP-BC intrasubunit interaction not previously observed) which they applied to a global analysis of conformational and catalytic states that offer some new structural snapshots of pyruvate carboxylase. The manuscript is conservatively titled and written: the manuscript explores PC structure by making new observations and drawing interesting correlations but it does not attempt to use these observations to offer substantive new explanations for how pyruvate carboxylase structure informs on the function of the enzyme. The discussion of allostery, for example, describes changes in angle that center on the CT-BC hinge at the allosteric domain, but falls short of proposing any new models for allosteric regulation (one of the major remaining questions in the function of this important enzyme; see "major revisions #5, below). Consequently, the manuscript undersells itself, leaving the reader to do much of the hard work of drawing out larger implications and possible mechanisms. There are valuable new observations in this work that will be of interest to those who study pyruvate carboxylase and related enzymes and many of the observations are consistent with prior studies on this enzyme, lending greater support to recent interpretations of pyruvate carboxylase function. Consequently, I am enthusiastic about this work being published and presented in the best possible light. I recommend several revisions, both major and minor, to improve the appeal of this manuscript beyond a narrow readership and to maximize its impact for those immersed in the study of pyruvate carboxylase.

Major revisions:

- 1. It is surprising how little the current manuscript ties back into this group's prior cryo EM work on PC (references 22 and 32). Where these works are loosely cited, they are mostly just used to describe general features of PC. Yet, the prior cryo EM study of *Staphylococcus aureus* PC from this same group (albeit at lower resolution) offered a proposed catalytic pathway that suggested a much more coordinated and regulated series of transitions for the BCCP domain, where the top and bottom layers of the tetramer were fully offset and the BCCP domain transitioned through a cross-layer interaction with the exo binding site at the PT/allosteric domain. The prior models from this group are not mentioned in the current study and no attempt is made to reconcile these different conclusions (compare the conclusions from figure 7 in the current manuscript to the scheme outlined in figure 6 from reference 32, for example). An explanation of the different conclusions will need to go beyond "reference 32 was in *Staphylococcus aureus* and the current work is in *Lactococcus lactis*" unless there is a very good rationale for why these enzymes would differ so significantly. How has the model changed and how are the different conclusions reconciled?**
- 2. In general, there is no mention made of the exo binding site anywhere in the current manuscript. BCCP-exo binding site interactions are commonly observed in crystal structures of PC, including in the *Lactococcus lactis* PC structure where biotin was observed to occupy one of the exo binding sites (similar to what was reported in RePC (reference 17)). The BCCP-exo binding site interaction has been seen most prominently in SaPC. Yet no mention was made in the current manuscript of a search for or examination of this conformational state. This is especially noteworthy since a close examination of crystal structures suggest that the presence of the BCCP domain in the exo binding site may be what is responsible for shifting the equivalent to helix 827-840 in the CT domain funnel. The analysis presented in figure 5 does not consider the possible role of BCCP in the exo binding site when considering CT domain helix constrictions. If the constriction of the funnel is related to the presence of BCCP in the adjacent exo binding site, it would be most exciting to correlate the presence of BCCP in the adjacent exo binding site with the ligand occupancy of the CT active site. The exo-binding conformational state of PC has been left unexplored and unexplained in the current manuscript. It must be more carefully addressed and considered.**
- 3. While the authors generally do well to relate their observations to the PC literature**

(points 1 and 2 above notwithstanding), they do not relate the findings from their heat map in figure 7 to the long simmering question of half-of-the-sites reactivity in biotin-dependent enzymes. It is interesting to see a strong **positive** correlation in BC_react in figure 7D. This would appear to be inconsistent with past proposals for half-of-the-sites-reactivity centered on the BC domain. Instead, the strong negative correlation in BC_react appears to be mediated by the CT domain (Figure 7b). Given the unique and powerful nature of the reported observations, a clearer recognition of how the current work ties into half-the-sites-reactivity debate is warranted, with appropriate reference to the literature.

4. The manuscript currently devotes much space to describing structural details that have already been well described in *Lactococcus lactis* and other PC systems. I agree that the unique catalytically competent conformation of the BCCP-BC domain interaction deserves a good measure of attention in this manuscript, but the molecular descriptions should focus more clearly on the unique and new structural insights that have been gained in the current work instead of redescribing molecular interactions that have been well described elsewhere. For example, is there anything new about the binding interactions with MgADP/ATP that haven't been described elsewhere? Is there anything new about the molecular interactions in the CT domain active site that haven't been described elsewhere? If new and different, make the new insights come out more clearly. If not different than previous reports, I recommend against significant attention to describing what has already been described. Aside from the new BCCP-BC domain interaction, the power of this manuscript is not in the molecular details that have already been defined by x-ray crystallography. The power of this manuscript lies in the global analysis and correlations between catalytic states and conformational changes. By devoting valuable space to established molecular details, the manuscript detracts from the exciting new contributions that this study is making.

5. The manuscript describes observations with acetyl-CoA (activator), ci-diAMP (inhibitor) and both. This study makes the interesting observation that the BC-CT hinge at the allosteric domain appears to be the overriding factor in facilitating allosteric activation. But disappointingly, the manuscript goes no further than simply pointing out this observation of differences and loosely implies that there may be some importance to the hinge angle. Given all the conformational states that have been captured here, I would prefer to see the manuscript connect this observation to the global conformational changes associated with catalytic turnover to say something new (and at least a little bit bold) about the mechanism of allostery in PC.

Minor revisions:

1. Page 2: Note that urea carboxylase has been reclassified to include a large group of guanidine carboxylases. Most urea carboxylases appear to be guanidine carboxylases (See doi: [10.1021/acs.biochem.0c00537](https://doi.org/10.1021/acs.biochem.0c00537))
2. Page 3: "The BCCP domain is translocated from the BC layer of its own monomer to the CT domain of the opposite subunit in the same." This is the primary translocation pathway, but others have been identified.
3. Page 3, last sentence of second paragraph: The idea that translocation is "triggered" is part of a longstanding notion that carrier domain motion in pyruvate carboxylase is "induced" to move from one site to the next. There is little evidence to support that. More neutral wording should be used to accommodate the possibility that the carrier domain could, instead, freely translocate between active sites according to a dynamic equilibrium model. In fact, the results of this study seem much more consistent with a dynamic equilibrium than an induced conformational change.
4. Page 4: The mention of the B subdomain lid opening and closing should reference the initial paper from Thoden, Blanchard, Holden and Waldrop (doi: [10.1074/jbc.275.21.16183](https://doi.org/10.1074/jbc.275.21.16183))
5. Page 4: It is difficult to get a clear sense of how well biotin is represented in the density. Its position makes more sense compared to the 3TW6 structure. Nevertheless, given the differences, it would be helpful to know the degree to which this new position for biotin is supported. Neither the ureido oxygen nor the valerate side chain appear to be well supported by the map. A figure offering a closer look at exactly how well the

modeled position for biotin is defined by the map is necessary.

6. Page 5: As presented in figure 3, the support for MgADP in the BC_open conformation is unconvincing. It is surprising that ADP would remain bound in the BC_open conformation. It is likely that this is a weak average between empty and ADP-bound conformations. Further qualification and explanation is needed on this point.

7. Page 6: In the CT_react structure, is it clear that the BCCP domain is originating from the neighboring subunit? Please clarify in the manuscript

8. Page 7: It is not clear what is meant by "relative variations". Relative variations in what? Are these variations in the catalytic state or in the rotational transformations? If this variation is with respect to rotational transformations, figure 7 does not relay that information. Figure 7 reports a heat map of related catalytic states for the individual domains. I very much like Figure 7, but I do not see how it can be used to describe that the relative variations are "small and in the range of +/- 5%".

9. Page 12: This reaction would require pyruvate to proceed. It is not listed in the list of reactants.

10. In many instances "CoA" is used as an abbreviation for acetyl CoA. This is confusing because Coenzyme A also activates PC and has been crystallized with PC (PDB 3HO8). Recommend that it be abbreviated as AcCoA.

With highest regards,
Martin St. Maurice

Reviewer #2 (Remarks to the Author):

The manuscript describes a detailed exploration of the conformational states of pyruvate carboxylase from *Lactococcus lactis* by single-particle cryo-EM. It describes a number of structures, including one in which the biotin-carboxyl carrier protein engages with the biotin carboxylase domain. The authors describe the effect of acetyl-CoA and c-di-AMP on the angle between biotin carboxylase and carboxyl transferase domains. They explore correlations between the conformational spectrum in adjacent units within the tetrameric complex. In general, the work appears to have been carried out with skill and care.

Some issues require further attention:

1. page 4, 'BC domain' section: 'This Mg²⁺ also coordinates the oxalate groups'; do the authors mean 'This Mg²⁺ is also coordinated by the carboxylate groups'?

2. It is not clear to me from the density shown in Fig. 3 that the local resolution allows unambiguous identification of ADP/ATP in this position. This is also indicated by the somewhat contradictory statements of the authors : 'The densities [...] show clear signal for ADP and ATP respectively' and, a few lines later : 'the density for ADP is well defined for the adenine and for the sugar backbones, however the region of the phosphates seems to be a mixture of different conformations'. The authors go further, claiming to resolve a difference in the strength of a salt bridge (no change in the distance of interaction mentioned). Based on the density shown, it is very difficult to confirm these claims.

3. The structure contains several ions at catalytically-relevant positions, which the authors have assigned as Mg²⁺, bicarbonate and Mn²⁺. The authors should either briefly summarize and cite the evidence for the assignment or, in cases where there is insufficient evidence, indicate to the reader that it represents a tentative assignment.

4. The authors used symmetry expansion to allow them to separate on the basis of conformational state, but it seems surprising that no reconstruction of the consensus refinement with D2 symmetry applied is provided. The description of the symmetry expansion sounds more complicated than necessary - are the authors aware that this is implemented in the Relion package? If they did use the Relion implementation, the further description and the self-citation (Lazaro et al 2021) appear to be unnecessary. If they used separate code, this should be made available. Finally, referring to the dataset as 'quadruplicated' is, in my opinion, less clear than referring to a symmetry-expanded particle set.

- 5. In Figure 1, why do the authors not show their own structure, rather than an existing crystal structure?**
- 6. In Figure 7, is it possible to include a scalebar relating the shade of red or blue to a quantitative estimate of the degree of correlation / anti-correlation?**

Smaller suggestions to the authors:

- 7. The first paragraph of the introduction spends quite a lot of time talking about single-particle cryo-EM and its application to problems of structural heterogeneity, which made this reviewer expect a paper about methods, whereas this paper seems to be about PC, and the methods applied are very standard. This section could probably be shortened and moved from the beginning of the intro to a later part.**
- 8. Is the L1PC + c-di-AMP dataset really carried out with the conditions stated in the methods? This would mean the use of around 100e/pix/s, which is far outside the ideal range for the Falcon III. The authors should check this.**
- 9. Some sentences are not correct as written, and the paper would benefit from careful proof-reading.**

This reviewer strongly encourages the authors to number lines for manuscript submission in the future, as also recommended by the journal. This saves on confusion for all involved!

RESPONSE TO REVIEWERS

Reviewer #1 (Remarks to the Author):

The manuscript by López-Alonso et al describes a series of conformational states in pyruvate carboxylase from *Lactococcus lactis* captured by unclassification techniques applied to cryo EM. Using this approach, this team has captured important new conformational states for pyruvate carboxylase (most notably, a catalytically competent BCCP-BC intrasubunit interaction not previously observed) which they applied to a global analysis of conformational and catalytic states that offer some new structural snapshots of pyruvate carboxylase. The manuscript is conservatively titled and written: the manuscript explores PC structure by making new observations and drawing interesting correlations but it does not attempt to use these observations to offer substantive new explanations for how pyruvate carboxylase structure informs on the function of the enzyme. The discussion of allostery, for example, describes changes in angle that center on the CT-BC hinge at the allosteric domain, but falls short of proposing any new models for allosteric regulation (one of the major remaining questions in the function of this important enzyme; see “major revisions #5, below). Consequently, the manuscript undersells itself, leaving the reader to do much of the hard work of drawing out larger implications and possible mechanisms. There are valuable new observations in this work that will be of interest to those who study pyruvate carboxylase and related enzymes and many of the observations are consistent with prior studies on this enzyme, lending greater support to recent interpretations of pyruvate carboxylase function. Consequently, I am enthusiastic about this work being published and presented in the best possible light. I recommend several revisions, both major and minor, to improve the appeal of this manuscript beyond a narrow readership and to maximize its impact for those immersed in the study of pyruvate carboxylase.

We thank the reviewer for the positive feedback about the contributions presented in the current manuscript. In general, we agree on the conservative nature of our interpretation. This is something that we have been discussing deeply and we thought that a clean presentation of the results and a cautious interpretation was a good approach. The large number of structures present in the current work and the correlations between them make the work difficult to communicate in a comprehensible way. However, since the reviewer encourages us to go a step (or several steps) further, we now include new notions throughout the discussion section (see below) on how LIPC works. Hope we have not gone too far this time...

Major revisions:

1. It is surprising how little the current manuscript ties back into this group’s prior cryo EM work on PC (references 22 and 32). Where these works are loosely cited, they are mostly just used to describe general features of PC. Yet, the prior cryo EM study of *Staphylococcus aureus* PC from this same group (albeit at lower resolution) offered a proposed catalytic pathway that suggested a much more coordinated and regulated series of transitions for the BCCP domain, where the top and bottom layers of the tetramer were fully offset and the BCCP domain transitioned through a cross-layer interaction with the exo binding site at the PT/allosteric domain. The prior models from this group are not mentioned in the current study and no attempt is made to reconcile these different conclusions (compare the conclusions from figure 7 in the current manuscript to the scheme outlined in figure 6 from reference 32, for example). An explanation of the different conclusions will need to go beyond “reference 32 was in *Staphylococcus aureus* and the current work is in *Lactococcus*

lactis” unless there is a very good rationale for why these enzymes would differ so significantly. How has the model changed and how are the different conclusions reconciled?

Sincerely, we do not know how to reconcile the current results of LIPC with our former SaPC study by cryoEM. There are several differences between these two works that make the comparison difficult.

Regarding the functioning of SaPC and LIPC, a big difference is the regulation by acetyl-CoA. LIPC does not need activation by acetyl-CoA and has a high intrinsic catalytic activity (as shown in figure 10), while SaPC and RePC need activation by acetyl-CoA.

The low-resolution maps for SaPC were understood using the crystallographic structures for SaPC (Yu et al., 2009) and RePC (Lietzan et al., 2011; St Maurice et al., 2007) tetramers that show different arrangements for PT/AL domains. These configurations are known as symmetric (SaPC) and asymmetric (RePC) respectively, and subunits differ in the angle between the CT and BC regions, where the PT/AL domain acts as the hinge. The asymmetric configuration has also been observed for *Listeria monocytogenes* PC (LmPC) (Sureka et al., 2014).

The asymmetric conformation is observed in the absence of the acetyl-CoA activator (LmPC) or in the presence of two molecules of acetyl-CoA (or ethyl-CoA) per tetramer (as in RePC), while the symmetric shows four molecules of CoA per tetramer (as in SaPC).

So, it is possible that the symmetric/asymmetric forms exist in PC that are sensitive to acetyl-CoA activation, and depending on the number of molecules bound to the PC tetramer in the PT/AL domain (the hinge in the structural change), the angle between the BC and CT domains changes and the oligomer maps different conformations.

The change of this angle is what we show in figure 10 for LIPC in the presence of allosteric regulators, but the maximum amplitude is about 10 degrees, while in two subunits of the asymmetric RePC this angle reaches 60-70 degrees. The fact that the different angles observed in PC using the PT/AL domain as the hinge are related to both elements, i.e., allosteric regulation and the possible symmetric/asymmetric transition is puzzling. We have not detected the asymmetric conformation in LIPC (and we have looked for it).

Of course, another difference is the state of the art of the cryoEM technique and the resolutions achieved in both works, around 11 Å for SaPC (published in 2014 with data from 2012-2013) and 2-3 Å for LIPC in the current work.

Furthermore, the structures for SaPC were produced with sets of around 14,000 particles and applying C2 symmetry, whereas the current LIPC data set includes approximately 340,000 tetramers and 1,360,000 subunits after symmetry expansion, which means that no symmetry is used at the tetramer level. In light of our current results, the use of C2 symmetry in the SaPC work does not seem like a good idea. In any case, our current cryoEM work is much better in all aspects.

Thus, do PC enzymes that require acetyl-CoA activation alternate between symmetric and asymmetric states depending on the number of acetyl-CoA molecules bound to the tetramer? Is the asymmetric configuration an active form? In such case, does the asymmetric tetramer follow a different pathway for the BCCP translocation? In the case of LIPC, the asymmetric state is not present because it is not sensitive to acetyl-CoA activation? We cannot answer these questions until we explore the active tetramers of RePC and/or SaPC at higher resolution. Meanwhile, we believe that introducing this topic in depth in the current

manuscript could be confusing and could obscure the findings of LIPC functioning. A different format (for instance, a review) might be a better place to speculate as we do not have a good answer.

Nevertheless, we have included a paragraph in the discussion that points out the absence of the asymmetric architecture in our data and some general considerations related to the asymmetric structures described previously (lines 392-407):

“In all the cryoEM maps for LIPC analyzed in this work, the subunits have a symmetrical organization without major differences in their overall architecture. This is relevant since an asymmetric configuration has been described in other PC tetrameric structures, such as in the crystallographic structures for RePC^{15,20}, and for *Listeria monocytogenes* PC (LmPC)³⁸, and in the low-resolution cryoEM structure for SaPC²³. In the asymmetric configuration, the position of the PT domain is significantly different when subunits from different layers are compared and there is a rotation of about 60-70° between BC and CT regions. In the cryoEM work of SaPC engaged in catalytic activity²³, although at low resolution, the asymmetric tetramer was understood as a catalytically relevant state. Our current results for LIPC do not show any asymmetric architecture and most of the relevant stages of the enzymatic activity are in display. The asymmetric quaternary structure has been described in PC enzymes with low intrinsic activity that require the activation by acetyl-coA (as it is the case for RePC, SaPC, and LmPC), and in tetrameric organizations without the activator³⁸ or with only two molecules of acetyl-CoA (or ethyl-CoA) bound to each tetramer^{15,20}. One possibility is that the asymmetric PC structure appears in tetramers that do not contain the four allosteric activator molecules that are needed to be fully competent. To clarify whether the asymmetric state has any enzymatic relevance, future studies are required.”

Lietzan, A.D., Menefee, A.L., Zeczycki, T.N., Kumar, S., Attwood, P.V., Wallace, J.C., Cleland, W.W., and St Maurice, M. (2011). Interaction between the biotin carboxyl carrier domain and the biotin carboxylase domain in pyruvate carboxylase from *Rhizobium etli*. *Biochemistry* 50, 9708-9723.

St Maurice, M., Reinhardt, L., Surinya, K.H., Attwood, P.V., Wallace, J.C., Cleland, W.W., and Rayment, I. (2007). Domain architecture of pyruvate carboxylase, a biotin-dependent multifunctional enzyme. *Science* 317, 1076-1079.

Sureka, K., Choi, P.H., Precit, M., Delince, M., Pensinger, D.A., Huynh, T.N., Jurado, A.R., Goo, Y.A., Sadilek, M., Iavarone, A.T., et al. (2014). The cyclic dinucleotide c-di-AMP is an allosteric regulator of metabolic enzyme function. *Cell* 158, 1389-1401.

Yu, L.P., Xiang, S., Lasso, G., Gil, D., Valle, M., and Tong, L. (2009). A symmetrical tetramer for *S. aureus* pyruvate carboxylase in complex with coenzyme A. *Structure* 17, 823-832.

2. In general, there is no mention made of the exo binding site anywhere in the current manuscript. BCCP-exo binding site interactions are commonly observed in crystal structures of PC, including in the *Lactococcus lactis* PC structure where biotin was observed to occupy one of the exo binding sites (similar to what was reported in RePC (reference 17)). The BCCP-exo binding site interaction has been seen most prominently in SaPC. Yet no mention was made in the current manuscript of a search for or examination of this conformational state. This is

especially noteworthy since a close examination of crystal structures suggest that the presence of the BCCP domain in the exo binding site may be what is responsible for shifting the equivalent to helix 827-840 in the CT domain funnel. The analysis presented in figure 5 does not consider the possible role of BCCP in the exo binding site when considering CT domain helix constrictions. If the constriction of the funnel is related to the presence of BCCP in the adjacent exo binding site, it would be most exciting to correlate the presence of BCCP in the adjacent exo binding site with the ligand occupancy of the CT active site. The exo-binding conformational state of PC has been left unexplored and unexplained in the current manuscript. It must be more carefully addressed and considered.

We have made dozens of classifications of the data set, and we have not detected any signal for the BCCP domain (even at very low-density threshold) next to the exo-binding site in LIPC. Since we have isolated a large number of catalytic states, if the exo binding site had any link to any of those states, it would have come up. Since we have not seen this exo site and its functional role is not clear, and the manuscript is already very dense, we would like to skip this issue.

3. While the authors generally do well to relate their observations to the PC literature (points 1 and 2 above notwithstanding), they do not relate the findings from their heat map in figure 7 to the long simmering question of half-of-the-sites reactivity in biotin-dependent enzymes. It is interesting to see a strong *positive* correlation in BC_{react} in figure 7D. This would appear to be inconsistent with past proposals for half-of-the-sites reactivity centered on the BC domain. Instead, the strong negative correlation in BC_{react} appears to be mediated by the CT domain (Figure 7b). Given the unique and powerful nature of the reported observations, a clearer recognition of how the current work ties into half-the-sites-reactivity debate is warranted, with appropriate reference to the literature.

Yes, it seems that the current results with LIPC disagree with many of the postulated mechanisms, including the half-of-the-sites model between layers through BC-BC communication (and the aforementioned transition between symmetric and asymmetric tetramer, and the exo binding site). We now include a specific discussion about the half-the-sites reactivity, where, if any, it goes through CT-CT contacts in the analyzed LIPC sample (lines 372-390):

“Noteworthy, the highest positive correlation corresponds to the BC_{react} pair joint by the BC-BC interaction surface (Fig. 7d), suggesting a cooperative behavior of BC regions in opposite layers to recruit biotin-carrying BCCP domains. This is also supported by the enrichment between BC_{react} and BC_{open}, and by the depletion between BC_{react} and BC_{closed} when the BC domains are in direct contact (Fig. 7d). All in all, it seems that the binding of BCCP to the BC domain in one layer stimulates the recruitment of the BCCP to the opposite layer when ready (BC_{closed}). This is in disagreement with the postulated negative cooperativity for biotin carboxylase dimers in the so-called half-of-sites reactivity mode, where BC subunits communicate to alternate in their catalytic states^{36,37}. This model is based on some observations of the alternate mode of action of the biotin carboxylase dimers of the ACC multienzyme complex³⁶. In our data with tetrameric LIPC, a clear negative correlation is observed between the BC domains of the subunits bound by their CT regions for the BC_{react} state (Fig. 7b). We have also observed that the PC subunits bound by their CT domains show a negative correlation in their catalytic state (Fig. 7b). The correlations between active sites and tetrameric motions are loose, suggesting that the movements of the BCCP domains linked by highly flexible loops are not tightly coupled to any of the conformational changes of the tetramer, but

that some of the motions allow or favor them. Nevertheless, there is a significant coupling between the CT-tilt motion and an asymmetric catalytic state of the CT active site between layers (Fig. 8). To decipher whether it is a genuine negative cooperativity mediated by CT-CT contacts and the possible mechanism for its regulation, additional insights are needed.”

4. The manuscript currently devotes much space to describing structural details that have already been well described in *Lactococcus lactis* and other PC systems. I agree that the unique catalytically competent conformation of the BCCP-BC domain interaction deserves a good measure of attention in this manuscript, but the molecular descriptions should focus more clearly on the unique and new structural insights that have been gained in the current work instead of redescribing molecular interactions that have been well described elsewhere. For example, is there anything new about the binding interactions with MgADP/ATP that haven't been described elsewhere? Is there anything new about the molecular interactions in the CT domain active site that haven't been described elsewhere? If new and different, make the new insights come out more clearly. If not different than previous reports, I recommend against significant attention to describing what has already been described. Aside from the new BCCP-BC domain interaction, the power of this manuscript is not in the molecular details that have already been defined by x-ray crystallography. The power of this manuscript lies in the global analysis and correlations between catalytic states and conformational changes. By devoting valuable space to established molecular details, the manuscript detracts from the exciting new contributions that this study is making.

We have been discussing this topic (detailed description of the CT active site) before submitting the manuscript. It is true that the atomic details of the CT site have been reported in several works. However, we think that since the current work reveals a complete sequence and not isolated snapshots, we should provide a detailed description. We have shortened the CT section referring to previously published works.

5. The manuscript describes observations with acetyl-CoA (activator), ci-diAMP (inhibitor) and both. This study makes the interesting observation that the BC-CT hinge at the allosteric domain appears to be the overriding factor in facilitating allosteric activation. But disappointingly, the manuscript goes no further than simply pointing out this observation of differences and loosely implies that there may be some importance to the hinge angle. Given all the conformational states that have been captured here, I would prefer to see the manuscript connect this observation to the global conformational changes associated with catalytic turnover to say something new (and at least a little bit bold) about the mechanism of allostery in PC.

This has also been an issue when writing the manuscript. The key point is how different angles at the PT/AL hinge change the conformational landscape of LIPC tetramers. As we stated in the manuscript, we suspect that this angle may modulate the accessibility of the BCCP domain to the CT active site (lines 411-412):

“The effect of this angle in PC activity could be mediated by the different accessibility of BCCP to the CT domain as observed in multibody refinement analysis.”

However, at this point we can only speculate, since for the structures compared in Figure 10 we only have data at the level of the conformational space for LIPC+AcCoA. The others are just

averages or crystallographic structures and there is no information about their motions. Furthermore, only the LIPC+AcCoA sample contains all the substrates and cofactors. Thus, we do not know which conformations are favored or disfavored in LIPC tetramers with different BC-CT angles.

However, we have included an advanced discussion about the possible mechanism of allosteric regulation (lines 412-425):

“We do not have access to the analysis of the motions that PC displays while bound to different allosteric modulators, but the clear correlation between the enzymatic activity and the angle that is shown in the hinge between the BC and CT domains (Fig. 10) suggests that allostery plays by narrowing the spectrum of available tetrameric motions.

In summary, we have resolved several catalytic states of LIPC with the allosteric activator acetyl-CoA and the motions that LIPC tetramers undergo during catalysis. By crossing these analyses, we conclude that the BCCP domain with the flexible linker can move between active sites driven by the affinity of carboxylated or non-carboxylated biotin and the readiness of the catalytic regions to accept it. The movement of the BCCP domain is not tightly coupled to the conformational changes of the tetramer, but the motions of the oligomer can facilitate its access to the active sites, and allosteric regulators modulate the landscape of these motions, that is, they narrow the conformational space which in turn conditions the functional space of the enzyme.”

Minor revisions:

1. Page 2: Note that urea carboxylase has been reclassified to include a large group of guanidine carboxylases. Most urea carboxylases appear to be guanidine carboxylases (See doi: 10.1021/acs.biochem.0c00537)

This dual role of UC is now included (lines 53-54):

“... and urea carboxylase (UC). Many of the UCs can also play as guanidine carboxylases⁷”

2. Page 3: “The BCCP domain is translocated from the BC layer of its own monomer to the CT domain of the opposite subunit in the same.” This is the primary translocation pathway, but others have been identified.

This sentence has been changed (lines 85-87):

“This pathway seems to be favored by the allosteric regulator acetyl-CoA, but the movement of each BCCP domain can follow different pathways, reaching the four active sites in its own layer²⁵.”

3. Page 3, last sentence of second paragraph: The idea that translocation is “triggered” is part of a longstanding notion that carrier domain motion in pyruvate carboxylase is “induced” to move from one site to the next. There is little evidence to support that. More neutral wording should be used to accommodate the possibility that the carrier domain could, instead, freely translocate between active sites according to a dynamic equilibrium model. In fact, the results

of this study seem much more consistent with a dynamic equilibrium than an induced conformational change.

This notion (related to major revision #5) is now included in the discussion section (lines 418-425):

“In summary, we have resolved several catalytic states of LIPC with the allosteric activator acetyl-CoA and the motions that LIPC tetramers undergo during catalysis. By crossing these analyses, we conclude that the BCCP domain with the flexible linker can move between active sites driven by the affinity of carboxylated or non-carboxylated biotin and the readiness of the catalytic regions to accept it. The movement of the BCCP domain is not tightly coupled to the conformational changes of the tetramer, but the motions of the oligomer can facilitate its access to the active sites, and allosteric regulators modulate the landscape of these motions, that is, they narrow the conformational space which in turn conditions the functional space of the enzyme.”

4. Page 4: The mention of the B subdomain lid opening and closing should reference the initial paper from Thoden, Blanchard, Holden and Waldrop (doi: 10.1074/jbc.275.21.16183)

Done with the new reference #26.

5. Page 4: It is difficult to get a clear sense of how well biotin is represented in the density. Its position makes more sense compared to the 3TW6 structure. Nevertheless, given the differences, it would be helpful to know the degree to which this new position for biotin is supported. Neither the ureido oxygen nor the valerate side chain appear to be well supported by the map. A figure offering a closer look at exactly how well the modeled position for biotin is defined by the map is necessary.

We have added a new panel in Figure 2 that zooms in the density for biotin.

6. Page 5: As presented in figure 3, the support for MgADP in the BC_{open} conformation is unconvincing. It is surprising that ADP would remain bound in the BC_{open} conformation. It is likely that this is a weak average between empty and ADP-bound conformations. Further qualification and explanation is needed on this point.

It is surprising to get the BC_{open} with ADP bound. To our knowledge, there is no previously described structure of an ATP-grasp domain with the open b-lid and with bound nucleotide (ATP or ADP). However, we are confident with the densities for ATP and ADP (new panels in Figure 3) which are quite strong and do not suggest low occupancy (as in a mixture of empty and bound ATP). A key point is the interaction of K116 with the β -phosphate, both in ATP (figure 3g) and in ADP (figure 3f) that “measures” the length of the tail of phosphates.

Another surprise is that we have not found an empty BC_{open} (with no nucleotide). One possibility is that a high concentration of ADP in the sample (after the initial reactions) could affect its release. In that case, we could expect a serious depletion of available ATP, but most of the BC classes show ATP bound. There are only about 10% of the BC domains with ADP (the BC_{open} class). Our interpretation is that under the conditions of our samples, the exchange

between ADP and ATP in the BC domain is very fast and moves rapidly the BC-lid to the closed state. In any case, we do not have a definitive answer.

7. Page 6: In the CT_{react} structure, is it clear that the BCCP domain is originating from the neighboring subunit? Please clarify in the manuscript

No, it is not clear. The loop connecting PT/AL to the BCCP domain is very well defined in the interaction with the BC region (as shown in Figure 1c) but it is not well resolved in the map for CT_{react}. The classification focused on the CT region included the realignment of the domain within a mask. Without this new alignment, the results were very poor, probably due to the large movements (tilting, opening and closing) of the CT catalytic site. In the final maps for complete tetramers after realignment of the CT region, the rest of the structure is blurred. We cannot trace the connection between the BCCP and PT/AL domains, but we do not know whether it is an effect of the realignment or whether it is a mixture of BCCP domains coming from different subunits. We have included a sentence (lines 202-204):

“In this CT_{react} complex, the density of the loop connecting BCCP to the PT domain is blurred, so the structure cannot define whether the BCCP domain comes from the opposite or from the same subunit.”

8. Page 7: It is not clear what is meant by “relative variations”. Relative variations in what? Are these variations in the catalytic state or in the rotational transformations? If this variation is with respect to rotational transformations, figure 7 does not relay that information. Figure 7 reports a heat map of related catalytic states for the individual domains. I very much like Figure 7, but I do not see how it can be used to describe that the relative variations are “small and in the range of +/- 5%”.

This was not clear in the previous version. We have now reworded this section clearly indicating that the heat map refers to the catalytic state of the reaction centers along the tetramer, and their deviations from the average of all subunits. We have also introduced some sentences focusing on the BC catalytic state of subunits bound by their BC-BC domains (related to major revision #3). Regarding the range of the variations (increase or decrease from the average number of particles in each subunit for a particular class), in the color scale (new) in Figure 7 the range is +/-10%, but only the BC_{react} classes are close to 10%. The rest of the panels have lower values in the former range of +/-5%. This was a mistake that now has been fixed (lines 223-227):

“In this way, we can record the increase or decrease of each class associated with a catalytic state in a specific position within the tetramer compared to the rest of the catalytic sites in the same or other subunits, building a heat map. The deviations from the average along the compared subunits were small and in the range of $\pm 10\%$, although most of the values are closer to $\pm 5\%$ (Fig. 7).”

The low values of the observed deviations could be explained by the mixture of tetramers involved in different stages and pathways. We look at pairs of catalytic sites, but we do not know what is going on in the other six reaction centers, and those may also affect the pair under scrutiny. With six catalytic centers and four possible states per center, the possible combinations are in the range of thousands (4^6). So, we can expect a lot of “noise” produced by the effect of other subunits. The heat map in Figure 7 shows most values close to zero, and only few significant deviations. We are confident about the significance of these numbers, since the behavior of the CT pairs fits very well with the correlation of the CT_{react} and CT_{empty} with the CT_{tilt} motion of the tetramer shown in Figure 8.

9. Page 12: This reaction would require pyruvate to proceed. It is not listed in the list of reactants.

Yes, this was a mistake. Pyruvate is now included.

10. In many instances "CoA" is used as an abbreviation for acetyl CoA. This is confusing because Coenzyme A also activates PC and has been crystallized with PC (PDB 3HO8). Recommend that it be abbreviated as AcCoA.

We have changed the abbreviation that was used in the figures for the sake of clarity.

With highest regards,
Martin St. Maurice

Reviewer #2 (Remarks to the Author):

The manuscript describes a detailed exploration of the conformational states of pyruvate carboxylase from *Lactococcus lactis* by single-particle cryo-EM. It describes a number of structures, including one in which the biotin-carboxyl carrier protein engages with the biotin carboxylase domain. The authors describe the effect of acetyl-CoA and c-di-AMP on the angle between biotin carboxylase and carboxyl transferase domains. They explore correlations between the conformational spectrum in adjacent units within the tetrameric complex. In general, the work appears to have been carried out with skill and care. Some issues require further attention:

1. page 4, 'BC domain' section: 'This Mg²⁺ also coordinates the oxalate groups'; do the authors mean 'This Mg²⁺ is also coordinated by the carboxylate groups'?

Right, this was a mistake and it has been fixed.

2. It is not clear to me from the density shown in Fig. 3 that the local resolution allows unambiguous identification of ADP/ATP in this position. This is also indicated by the somewhat contradictory statements of the authors : 'The densities [...] show clear signal for ADP and ATP respectively' and, a few lines later : 'the density for ADP is well defined for the adenine and for the sugar backbones, however the region of the phosphates seems to be a mixture of different conformations'. The authors go further, claiming to resolve a difference in the strength of a salt bridge (no change in the distance of interaction mentioned). Based on the density shown, it is very difficult to confirm these claims.

We are confident with the assignment of ADP and ATP in the cryoEM maps for the BC_closed and BC_open classes (related to minor revision #6 from reviewer 1), and now we show additional panels in Figure 3 (3f and 3g) displayed a high density threshold. In these panels, it can be seen that K116 interacts with the β -phosphate of both ADP and ATP. In the case of ADP, the distance from the sugar to the interaction point with K116 cannot accommodate three

phosphates. Nevertheless, it is true that there is a paradox between the adjective “clear” and the poorer definition of the phosphates in the density for ADP. Thus, we have now removed the “clear” statement.

Regarding the changes in the potential salt bridge between R341 and E239, the density for R341 in the BC_open class (Figure 3d) can fully accommodate the full side chain since the salt bridge with E239 restrains its movement. In the case of the BC_closed (Figure 3e), the density for R341 vanishes and most of the side chain remains outside of the cryoEM envelope. This suggests that R341 is now free from the interaction with E239. There is no a large movement of R341, just a change in the orientation. Furthermore, the changes in this salt bridge are supported by previous work as referenced in the text. We have reformulated this section (lines 157-161):

“In the presence of MgATP, this loop binds to the ribose of the nucleotide, the density of the salt bridge between Glu239 and Arg341 weakens, and the density of Arg341 changes its orientation and cannot accommodate the full side chain of the amino acid, suggesting that Arg341 is more free from the interaction and that it can map the configuration required to receive an incoming biotin.”

3. The structure contains several ions at catalytically-relevant positions, which the authors have assigned as Mg²⁺, bicarbonate and Mn²⁺. The authors should either briefly summarize and cite the evidence for the assignment or, in cases where there is insufficient evidence, indicate to the reader that it represents a tentative assignment.

Cryo-EM gives evidence of the position of these cations at this resolution but not their unambiguous identity. The positions and identities of Mg²⁺ and bicarbonate in the BC active site have been widely described in previous works (Chou et al., 2009). In the case of the CT active site, Mn²⁺ has been described for crystallographic models of *S. aureus* (Yu et al., 2013), *L. lactis* (Choi et al., 2017), *L. monocytogenes* (Sureka et al., 2014) and *H. sapiens* (Xiang & Tong, 2008) PC. On the other hand, the PC structure of *R. etli* (st. Maurice et al., 2007) contains a Zn²⁺. We have used Mn²⁺ in our models to be consistent with previous works. Since we are not making an issue about the ions, and there are already many topics in the manuscript, we think that we can leave the assignment based on previous structures.

Choi, P. H., Vu, T. M. N., Pham, H. T., Woodward, J. J., Turner, M. S., & Tong, L. (2017). Structural and functional studies of pyruvate carboxylase regulation by cyclic di-AMP in lactic acid bacteria. *Proceedings of the National Academy of Sciences of the United States of America*, 114(35), E7226–E7235. <https://doi.org/10.1073/pnas.1704756114>

Chou, C.-Y., Yu, L. P. C., & Tong, L. (2009). Crystal structure of biotin carboxylase in complex with substrates and implications for its catalytic mechanism. *The Journal of Biological Chemistry*, 284(17), 11690–11697. <https://doi.org/10.1074/jbc.M805783200>

st. Maurice, M., Reinhardt, L., Surinya, K. H., Attwood, P. v., Wallace, J. C., Cleland, W. W., & Rayment, I. (2007). Domain architecture of pyruvate carboxylase, a biotin-dependent multifunctional enzyme. *Science*, 317(5841), 1076–1079. <https://doi.org/10.1126/science.1144504>

Sureka, K., Choi, P. H., Precit, M., Delince, M., Pensinger, D. A., Huynh, T. N., Jurado, A. R., Goo, Y. A., Sadilek, M., Iavarone, A. T., Sauer, J.-D., Tong, L., & Woodward, J. J. (2014). The cyclic dinucleotide c-di-AMP is an allosteric regulator of metabolic enzyme function. *Cell*, 158(6), 1389–1401. <https://doi.org/10.1016/j.cell.2014.07.046>

Xiang, S., & Tong, L. (2008). Crystal structures of human and *Staphylococcus aureus* pyruvate carboxylase and molecular insights into the carboxyltransfer reaction. *Nature Structural & Molecular Biology*, 15(3), 295–302. <https://doi.org/10.1038/nsmb.1393>

Yu, L. P. C., Chou, C.-Y., Choi, P. H., & Tong, L. (2013). Characterizing the Importance of the Biotin Carboxylase Domain Dimer for *Staphylococcus aureus* Pyruvate Carboxylase Catalysis. *Biochemistry*, 52(3), 488–496. <https://doi.org/10.1021/bi301294d>

4. The authors used symmetry expansion to allow them to separate on the basis of conformational state, but it seems surprising that no reconstruction of the consensus refinement with D2 symmetry applied is provided. The description of the symmetry expansion sounds more complicated than necessary - are the authors aware that this is implemented in the Relion package? If they did use the Relion implementation, the further description and the self-citation (Lazaro et al 2021) appear to be unnecessary. If they used separate code, this should be made available. Finally, referring to the dataset as 'quadruplicated' is, in my opinion, less clear than referring to a symmetry-expanded particle set.

The symmetry expansion was done before Relion implementation with a home-made script that we have now included in the submission. In addition, with our script we can keep track of each single particle and its rotations. This information is what we use for the study of correlation between classes. Thus, we use our own script for symmetry expansion, the same used by Lazaro et al 2021. We have uploaded the script used for symmetry expansion and the one used for the correlation between classes.

5. In Figure 1, why do the authors not show their own structure, rather than an existing crystal structure?

The idea in Figure 1 is to show that the BCCP mobile domain is not observed in the average of our cryoEM data. We think that it might be difficult for a reader to understand an atomic model coming from the cryoEM data and including the BCCP region before the classifications.

6. In Figure 7, is it possible to include a scalebar relating the shade of red or blue to a quantitative estimate of the degree of correlation / anti-correlation?

This scale bar is now included in Figure 7.

Smaller suggestions to the authors:

7. The first paragraph of the introduction spends quite a lot of time talking about single-particle cryo-EM and its application to problems of structural heterogeneity, which made this reviewer expect a paper about methods, whereas this paper seems to be about PC, and the methods applied are very standard. This section could probably be shortened and moved from the beginning of the intro to a later part.

It is true that we have not developed a new method. Actually, the home-made symmetry expansion might have been a new tool, but now it is implemented at least in Relion. However, we think that our approach is novel at the level of analysis of bottom-up (catalytic centers) and up-bottom (general motions) heterogeneity, and the correlations between these two sets. To our knowledge, the strategy is novel and seems well fitted to study multi-pathway reactions that occur in oligomeric enzymes such as PC.

The key point in that section of the introduction is the sentence (lines 43-44):

“The question is how far cryoEM can go in deciphering the functional structures of oligomeric complexes involved in multi-pathway reactions.”

And we would like to keep this section as it is.

8. Is the L1PC + c-di-AMP dataset really carried out with the conditions stated in the methods? This would mean the use of around 100e/pix/s, which is far outside the ideal range for the Falcon III. The authors should check this.

The reviewer is right about the high dose. It was our mistake when we sent samples to different EM platforms and different direct detectors, and we used mixed setups. Unfortunately, this was not corrected by the operator of the microscope. Therefore, we are aware of the poorer performance of the camera under these conditions, but the resulting cryoEM map is still good and, of course, the dose weighting during motion correction limits the contribution of images from damaged particles.

9. Some sentences are not correct as written, and the paper would benefit from careful proof-reading.

The manuscript has been revised by an English native.

This reviewer strongly encourages the authors to number lines for manuscript submission in the future, as also recommended by the journal. This saves on confusion for all involved!

Right. Sorry we missed this.

Reviewer #1 (Remarks to the Author):

The authors have extended their discussion and analysis in order to comment on features such as positive/negative cooperativity in the BC domain, symmetry/asymmetry and allostery. I recognize and respect that the authors are uncomfortable going too far in postulating mechanisms based on the current data. I believe the paper now strikes a fine balance between description, analysis and postulation. There are many interesting and well-presented findings in this manuscript. I recommend the current manuscript for publication.

Reviewer #2 (Remarks to the Author):

Most concerns have been addressed. I reiterate the following point.

Regarding metal ion identification: I am perfectly happy to accept that the identity of the metal ions is not crucial to the conclusions of the manuscript and that they have been (tentatively) assigned according to previous assignments. What I find unacceptable, as I said in my original review, is the continued use of statements in the revised version that suggest to the reader that the identity of these ions is known rather than assumed, such as:

Lines 212-213: This region contains a Mg²⁺ ion bound to the main chain of residues Val519 and Thr522, and the side chains of Asp753 (Suppl. Fig. 3c). This ion has been observed in RePC20, however, its role is not clear¹⁸.

Since neither the density nor any other evidence in this study specifically support assignment of the ion as Mg²⁺, it is simple good practice to be honest about this to readers (some of whom are not structural biologists and have no idea what information was available for this assignment) by writing, for example

This region contains an ion, tentatively assigned as Mg²⁺ on the basis of a similar assignment in RePC20, bound to the main chains ...'

It is entirely possible that, at some point in the future, someone WILL care what ion is present in this position. They will see your paper and reference 20, and may believe that these represent independent observations of Mg²⁺ in this position. You owe it to that reader to be honest about your how much information your experimental data provide.

An additional point:

The modelling of ADP in the BCopen state should be checked (7ZZ5). The orientation of the adenine in this structure is flipped relative to its position in pdb 3G8C and its position in ATP-bound structures (this study and others).

Response to reviewers

Reviewer #1 (Remarks to the Author):

The authors have extended their discussion and analysis in order to comment on features such as positive/negative cooperativity in the BC domain, symmetry/asymmetry and allostery. I recognize and respect that the authors are uncomfortable going too far in postulating mechanisms based on the current data. I believe the paper now strikes a fine balance between description, analysis and postulation. There are many interesting and well-presented findings in this manuscript. I recommend the current manuscript for publication.

We thank the reviewer for this positive feedback. We acknowledge that the review process has improved the manuscript.

Reviewer #2 (Remarks to the Author):

Most concerns have been addressed. I reiterate the following point.

Regarding metal ion identification: I am perfectly happy to accept that the identity of the metal ions is not crucial to the conclusions of the manuscript and that they have been (tentatively) assigned according to previous assignments. What I find unacceptable, as I said in my original review, is the continued use of statements in the revised version that suggest to the reader that the identity of these ions is known rather than assumed, such as:

Lines 212-213: This region contains a Mg²⁺ ion bound to the main chain of residues Val519 and Thr522, and the side chains of Asp753 (Suppl. Fig. 3c). This ion has been observed in RePC20, however, its role is not clear¹⁸.

Since neither the density nor any other evidence in this study specifically support assignment of the ion as Mg²⁺, it is simple good practice to be honest about this to

readers (some of whom are not structural biologists and have no idea what information was available for this assignment) by writing, for example

This region contains an ion, tentatively assigned as Mg^{2+} on the basis of a similar assignment in RePC20, bound to the main chains ...'

It is entirely possible that, at some point in the future, someone WILL care what ion is present in this position. They will see your paper and reference 20, and may believe that these represent independent observations of Mg^{2+} in this position. You owe it to that reader to be honest about your how much information your experimental data provide.

We now see the point raised by the reviewer. Accordingly, we clearly state along the manuscript that the identities of metal ions are based on previous works. This way new sentences (in blue in the manuscript) are included:

-lines 129-131

Although at lower occupancy, there is a density consistent with a second cation coordinating the ATP γ -phosphate (asterisks in Fig. 2b,c) which has been proposed to be a second Mg^{2+} ion²⁷

-lines 196-202

The active site contains a cation, which is coordinated to His732, His734, Arg533, Asp534, the two carboxylic atoms of carboxylsine 703 and a water molecule that also interacts with pyruvate O3. This ion has been identified as Mn^{2+} for the crystallographic models of *S. aureus*¹³, *L. lactis*³⁰, *L. monocytogenes*³¹ and human¹⁴ PC. On the other hand, the structure of PC from *R. etli*²⁰ contains a Zn^{2+} . We have assigned it as Mn^{2+} in our models to be consistent with previous works. The arrangement of the active site of CT agrees with that proposed by Sheng and coworkers³².

-lines 221-223

This region contains a cation bound to the main chains of residues Val519 and Thr522, and the side chain of Asp753 (Suppl. Fig. 3c). This ion has been tentatively assigned as Mg^{2+} based on a similar assignment in RePC²⁰, however, its role is not clear¹⁸.

An additional point:

The modelling of ADP in the BC_{open} state should be checked (7ZZ5). The orientation of the adenine in this structure is flipped relative to its position in pdb 3G8C and its position in ATP-bound structures (this study and others).

The reviewer is right, the orientation of ADP in the BC_{open} is flipped. This is the result of a rigid body fitting into the density. We have re-evaluated the statistics of the density maps for BC_{open} and BC_{closed} using smaller boxes. Rendering of the maps at 2.5 σ density thresholds gives a good representation of the protein in both maps. The density for ATP is also visible at the same threshold. However, the density for ADP needs to be rendered at 1.5 σ to account for the entire molecule. Based on that we understand that the occupancy of ADP is lower than the occupancy of ATP and compatible with a partial release of the nucleotide in the BC_{open} class (which makes sense). Considering the lower occupancy and the poorer definition of the density we have removed the atomic model for ADP from the modeled atomic structure (7ZZ5) and the representation of ADP from figures 3b, 3d, 3f and 4b.

This is now explained in the main text (lines 144-166):

Comparing the maps for BC_{open} and BC_{closed} states (Fig. 3) it is seen that the B-subdomain lid in BC_{open} opens around 50° (Fig. 3a, and Suppl. Movie M1) what could allow the release of ADP and the entrance of new substrate. The residues forming the hinge of the molecular lid are Ile131 and Glu203. Arg171 forms a new salt bridge with Glu203 that could stabilize the open conformation of the B-subdomain lid (Fig. 3b). There is a strong density for ATP in the BC_{closed} state (rendered at 2.5 σ in Fig. 3g), but the map for the BC_{open} class shows a weaker density in the nucleotide binding site (visible at 1.5 σ in Fig. 3f) compatible with partial release of the nucleotide upon opening of the B-subdomain lid. We attribute this density to unreleased ADP molecules present in some of the processed particles. Essentially, the interaction of Lys116 with the β -phosphate of ATP in the BC_{closed} state (Fig. 3g) seems to be conserved in the BC_{open} class between the same amino acid and the last phosphate of the putative ADP molecule (Fig. 3f). In this configuration, the ADP would interact only with the amine of Lys116 and the main chain

of Ile202 at both ends of the molecule, and the side interactions are missing. By comparison of the interaction networks in the BC_{open} (Fig. 3b,d) and in the BC_{closed} (Fig. 3c,e), it can be seen that the interactions of MgATP with Glu274, Glu286, His207, Gln231, Lys120 and the T-loop are lost upon the opening of the B-subdomain lid. The missing interactions in the BC_{open} between the nucleotide and the 231-239 loop liberate this region of the BC domain and Glu239 can interact with Arg341 through a salt bridge (Fig. 3d). In the presence of MgATP, this loop binds to the ribose of the nucleotide, the density of the salt bridge between Glu239 and Arg341 weakens, and the density of Arg341 changes its orientation and cannot accommodate the full side chain of the amino acid, suggesting that Arg341 is more free from the interaction and that it can map the configuration required to receive an incoming biotin. This salt bridge between Arg341 and Glu239 has been reported to be produced only after ATP cleavage to avoid the reentry of carboxybiotin into the active site and prevent its decarboxylation²⁸.